# Circuit-based intervention corrects excessive dentate gyrus output in the fragile X mouse model

Pan-Yue Deng[1], Ajeet Kumar[2], Valeria Cavalli[2], Vitaly A Klyachko[1]*

[1]Department of Cell Biology and Physiology, Washington University School of Medicine, St Louis, United States; [2]Department of Neuroscience, Washington University School of Medicine, St Louis, United States

**Abstract** Abnormal cellular and circuit excitability is believed to drive many core phenotypes in fragile X syndrome (FXS). The dentate gyrus is a brain area performing critical computations essential for learning and memory. However, little is known about dentate circuit defects and their mechanisms in FXS. Understanding dentate circuit dysfunction in FXS has been complicated by the presence of two types of excitatory neurons, the granule cells and mossy cells. Here we report that loss of FMRP markedly decreased excitability of dentate mossy cells, a change opposite to all other known excitability defects in excitatory neurons in FXS. This mossy cell hypo-excitability is caused by increased Kv7 function in *Fmr1* knockout (KO) mice. By reducing the excitatory drive onto local hilar interneurons, hypo-excitability of mossy cells results in increased excitation/inhibition ratio in granule cells and thus paradoxically leads to excessive dentate output. Circuit-wide inhibition of Kv7 channels in *Fmr1* KO mice increases inhibitory drive onto granule cells and normalizes the dentate output in response to physiologically relevant theta–gamma coupling stimulation. Our study suggests that circuit-based interventions may provide a promising strategy in this disorder to bypass irreconcilable excitability defects in different cell types and restore their pathophysiological consequences at the circuit level.

*For correspondence:
klyachko@wustl.edu

Competing interest: The authors declare that no competing interests exist.

## eLife assessment

This is a **fundamental** work that significantly advances our understanding of the role of mossy cells in the dentate gyrus in fragile X syndrome. The carefully designed and executed extensive series of experiments provide **compelling** evidence that changes in their excitability occur due to upregulation of Kv7 currents. The study unveils the underlying mechanisms of the disease, and therefore the work will be of interest to neuroscientists working on various aspects of fragile X pathology. In addition, it also provides insights into how neuronal activity is balanced in networks through diverse cellular mechanisms.

## Introduction

Fragile X syndrome (FXS) is the leading monogenic cause of intellectual disability and autism. This disorder arises from mutations in the *Fmr1* gene resulting in a loss of fragile X messenger ribonucleoprotein (FMRP) (*Salcedo-Arellano et al., 2020*; *Willemsen and Kooy, 2017*). Hippocampus, the brain area implicated as a central hub for learning and memory, is one of the most affected brain regions in FXS (*Bostrom et al., 2016*). Within the hippocampus, dentate gyrus receives the bulk of cortical inputs and plays a critical role in many core information processing functions (*Amaral et al., 2007*), which are often dysregulated in neurodevelopmental disorders, including FXS (*Deng et al., 2022*; *Eadie et al.,*

*2012*; *Ghilan et al., 2015*; *Lee et al., 2020*; *Yau et al., 2018*; *Yau et al., 2016*; *Yun and Trommer, 2011*). Distinct from other hippocampal regions, dentate gyrus has two types of glutamatergic excitatory neurons (*Figure 1A*): granule cells and mossy cells (MCs) (*Amaral et al., 2007*). Granule cells are the first-station neurons of canonical trisynaptic pathway and the only type of output cells in the dentate gyrus (*Amaral et al., 2007*), whereas MCs comprise a significant portion of neurons in the hilus region (*Scharfman, 2016*; *Scharfman, 2018*), which is sandwiched between the two blades of the granule cell layer. MCs are in a unique position to control dentate gyrus output because these cells not only can directly excite granule cells, but also indirectly inhibit them through innervating local GABAergic interneurons (*Amaral et al., 2007*), which play a central role in controlling granule cell activity (*Pelkey et al., 2017*). Both MCs and hilar interneurons have been implicated in various pathological conditions (*Pelkey et al., 2017*; *Scharfman, 2016*; *Scharfman, 2018*; *Noebels et al., 2012*). However, even though dentate gyrus has attracted extensive attention in the FXS field (*Deng et al., 2022*; *Lee et al., 2020*; *Modgil et al., 2019*; *Monday et al., 2022*; *Remmers and Contractor, 2018*; *Sathyanarayana et al., 2022*; *Yau et al., 2018*), whether or not MC function is abnormal and contribute to hippocampal circuit abnormalities in FXS remains unexplored.

Abnormal neural excitability in FXS has been often linked to dysregulated expression and/or function of various ion channels (*Deng and Klyachko, 2021*), and particularly the family of K$^+$ channels, which control neuronal excitability by regulating the action potential (AP) threshold, duration, frequency, and firing patterns. Kv7 (KCNQ) channels are a subthreshold-activated and non-inactivating K$^+$ conductance that plays key roles in regulating neuronal excitability throughout the brain (*Brown and Passmore, 2009*; *Delmas and Brown, 2005*; *Jones et al., 2021*; *Liu et al., 2021*), including hippocampal neurons (*Gu et al., 2005*; *Incontro et al., 2021*; *Martinello et al., 2015*; *Peters et al., 2005*). Increasing evidence suggests that Kv7 channels contribute to synaptic plasticity, learning/memory, and behavior (*Baculis et al., 2020*), and their dysfunction contributes to a variety of neurodevelopmental disorders (*Baculis et al., 2020*; *Gilling et al., 2013*; *Jentsch, 2000*; *Jones et al., 2021*; *Liu et al., 2021*; *Miceli et al., 2015*; *Miceli et al., 2008*; *Nappi et al., 2020*; *Peters et al., 2005*; *Springer et al., 2021*). Two members of the Kv7 family (Kv7.2 and Kv7.3) have been identified as targets of FMRP's translational regulation (*Darnell et al., 2011*), yet it is unknown whether Kv7 expression/function is affected by FMRP loss and plays a role in pathophysiology of FXS.

Here, we report an unexpected hypo-excitability of MCs in FXS mouse model, resulting from upregulation of Kv7 function. We uncover the implications of this defect to dentate dysfunction at the synaptic, cellular, and circuit levels. While this MC excitability defect is unique in being opposite in direction to all other glutamatergic neurons in FXS mouse model, it nevertheless paradoxically leads to an elevated excitation/inhibition (E/I) ratio in granule cells and abnormal circuit processing in the dentate circuit via reduced activity of local inhibitory networks. Our analyses using circuit-wide inhibition of Kv7 channels in *Fmr1* knockout (KO) mice suggest that circuit-based interventions may represent a promising strategy to bypass the need to correct irreconcilable excitability defects in different cell types and restore their pathophysiological consequences in this disorder.

## Results

### Decreased excitability of dentate MCs in the *Fmr1* KO mice

MCs comprise a large fraction of the cells in the dentate hilar region and are implicated in multiple pathological conditions (*Scharfman, 2016*; *Scharfman, 2018*). To investigate whether MCs are involved in the pathophysiology of FXS, we first examined the excitability of MCs in wildtype (WT) and *Fmr1* KO mice, the FXS mouse model. APs were recorded from MCs at different membrane potentials (from –64 to –55 mV, set by constant current injection). Unexpectedly, we found that loss of FMRP significantly decreased excitability of MCs, as evident by the reduced number of APs fired (*Figure 1B and C*; statistical data for every measurement in this study are listed in *Supplementary file 1*). This hypo-excitability is opposite to the changes previously observed in other excitatory neurons in FXS models, which typically exhibit hyperexcitability (*Contractor et al., 2015*; *Deng and Klyachko, 2021*). To verify this observation, we employed a ramp protocol to evoke APs and examined various AP parameters (*Deng et al., 2021*; *Deng et al., 2019*; *Deng et al., 2022*). Similarly, we found a decreased number of APs fired (*Figure 1D and E*), as well as increased voltage threshold, rheobase, and rheobase charge transfer in MCs of *Fmr1* KO mice (*Figure 1D and F–H*), confirming the hypo-excitable

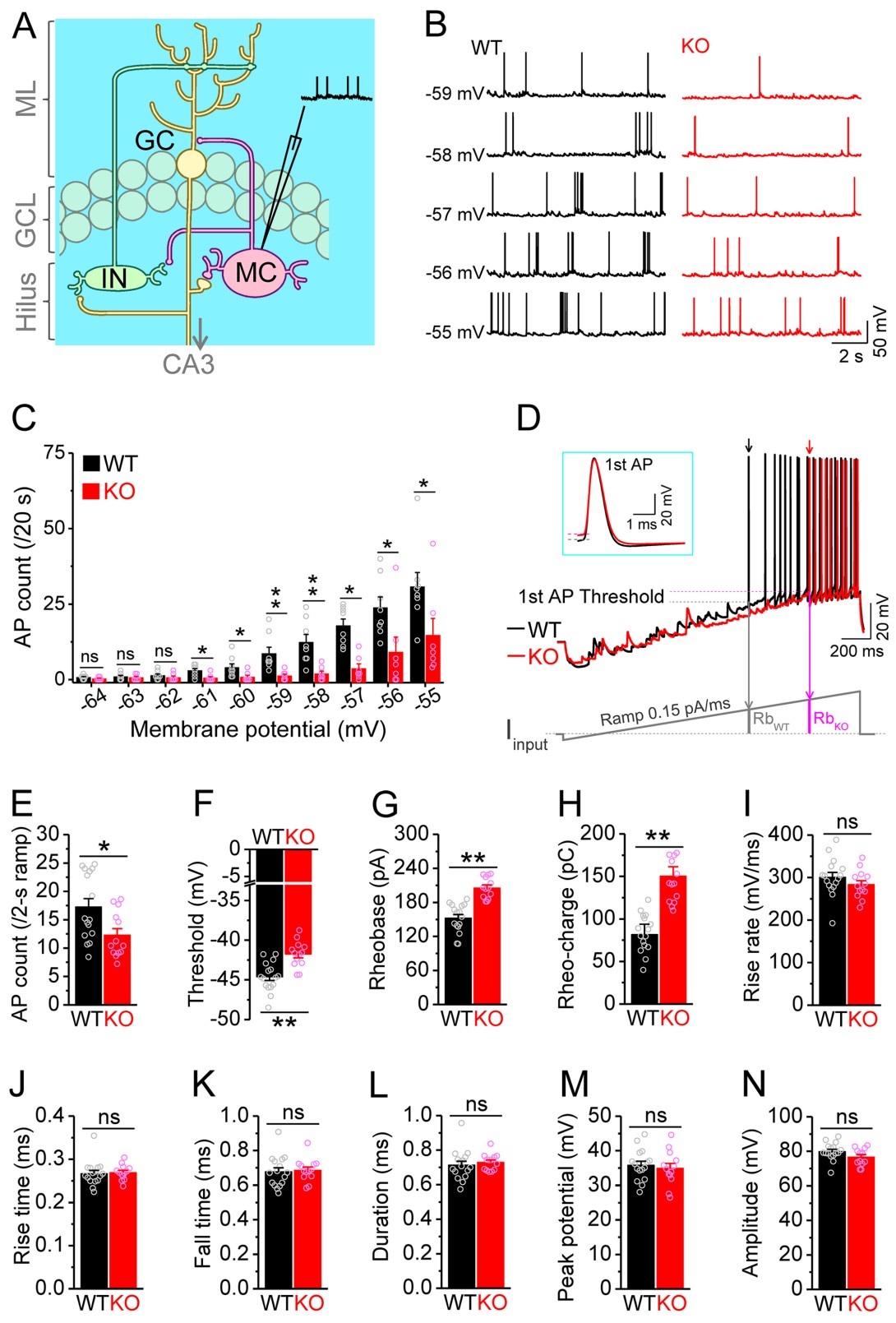

**Figure 1.** Decreased excitability of dentate mossy cells (MCs) in *Fmr1* knockout (KO) mice. (**A**) Schematic illustration of dentate circuit organization and recordings of action potentials (APs) from MCs. Note, for simplicity and clarity, we only show one type of interneurons (IN) with axons terminating onto distal dendrites of a granule cell (GC). The arrow indicates axons of GCs projecting to CA3. ML, molecular layer; GCL, granule cell layer. (**B**) Sample traces of spontaneous APs recorded at different membrane potentials from MCs in wildtype (WT) (black) and *Fmr1* KO (red) mice. (**C**) Summary data

*Figure 1 continued on next page*

Figure 1 continued

for experiments exemplified in (**B**) showing decreased number of APs at membrane potentials of –61 through –55 mV in KO MCs. Scatter circles indicate individual data points for this and all subsequent bar graphs in this study. (**D**) Determination of AP threshold and rheobase by a ramp current injection (lower trace, ramp rate 0.15 pA/ms). Only the first APs (arrows, which were expanded and aligned by the time of threshold in inset) were used to estimate AP parameters. The horizontal lines (inset) indicate threshold of the first APs. In the lower panel, $Rb_{WT}$ and $Rb_{KO}$ denote rheobase current intensity at threshold time point, and the area (integrating time and input current) enclosed by dotted line, current ramp and $Rb_{WT}$ (or $Rb_{KO}$) are rheobase charge transfer. (**E–H**) Summary data showing decreased number of APs during 2 s ramp (**E**), increased voltage threshold (**F**), rheobase (**G**), and rheobase charge transfer (**H**) in KO MCs. (**I–N**) AP upstroke maximum rise rate (**I**), rise time (**J**), fall time (**K**), duration (**L**), peak potential (**M**), and amplitude (**N**). *p<0.05; **p<0.01; ns, not significant. The statistical data are listed in ***Supplementary file 1***. Data are mean ± SEM.

state of MCs in the absence of FMRP. There were no significant changes in AP maximum rise rate, rise time, fall time, duration, peak potential, and amplitude in KO MCs (***Figure 1I–N***), indicating that the transient $Na^+$ current and fast activating $K^+$ conductances are likely unaffected in MCs of *Fmr1* KO mice.

The MC hypo-excitability in *Fmr1* KO mice can be attributed to cell-autonomous and/or circuit defects. We first asked whether this defect was caused by changes in the intrinsic passive membrane properties of MCs by examining the resting membrane potential (RMP), cell capacitance, and input resistance at RMP level, and found all of them to be unaffected in *Fmr1* KO mice (***Figure 2A–C***). However, since input resistance around the threshold level contributes directly to AP initiation, we examined it again at –45 mV (close to threshold level) using a depolarization step in the pharmacologically isolated MCs (using a combination of blockers against both glutamate and GABA receptors [in µM]: 10 NMDA, 50 APV, 10 MPEP, 5 gabazine and 2 CGP55845) (***Deng et al., 2019***; ***Deng et al., 2022***), as well as in the presence of 1 µM tetrodotoxin (TTX) and 10 µM $CdCl_2$ to block $Na^+$ and $Ca^{2+}$ channels, respectively. Under these conditions, the input resistance was significantly lower in *Fmr1* KO than WT MCs (***Figure 2D and E***), suggesting that the reduced input resistance around the threshold potential may be a primary contributor to MC hypo-excitability in *Fmr1* KO mice.

## Enhanced Kv7 function causes MC hypo-excitability in *Fmr1* KO mice

Among conductances active at sub-threshold potentials, Kv7 channels are known to play a major role in determining the input resistance (***Baculis et al., 2020***; ***Greene and Hoshi, 2017***; ***Jones et al., 2021***; ***van der Horst et al., 2020***). We thus tested whether the reduced input resistance in *Fmr1* KO MCs is associated with Kv7 dysfunction. We found that the Kv7 blocker XE991 (10 µM) increased input resistance in all tested MCs from both WT and *Fmr1* KO (***Figure 2E***), with a significantly larger effect in KO mice (***Figure 2F***). Importantly, XE991 abolished the difference in input resistance at –45 mV between WT and *Fmr1* KO cells (***Figure 2E***). Moreover, Kv7 inhibition with XE991 also caused a larger shift in the holding current in KO than WT MCs (***Figure 3A***), and a larger membrane depolarization in KO than WT MCs when the membrane potential was initially set at –45 mV (***Figure 3B***). We further observed that loss of FMRP increased Kv7 currents in MCs as measured by a ramp protocol (***Figure 3C and D***). Together, these results strongly suggest that Kv7 function is increased in *Fmr1* KO MCs.

The 'classical' Kv7 current is carried predominantly by heteromeric KCNQ2/KCNQ3 channels (***Brown and Passmore, 2009***). KCNQ2 and KCNQ3 have been identified as targets of FMRP translational regulation (***Darnell et al., 2011***) and are highly expressed in the hippocampus (***Tzingounis et al., 2010***). We thus examined whether KCNQ2 and KCNQ3 levels were altered by FMRP loss, but found no changes in expression of either isoform in the whole brain lysate as measured by western blot (***Figure 3—figure supplement 1A and B***). To exclude the possibility that expression variation among brain regions masked KCNQ changes in a particular area, we performed the same experiment on dentate gyrus lysate only. We found no detectable changes in expression of KCNQ2 or KCNQ3 in *Fmr1* KO mice (***Figure 3—figure supplement 1C and D***). We did not further investigate the precise mechanisms underlying enhancement of Kv7 function in the absence of FMRP since this study primarily focuses on the functional consequences of abnormal cellular and circuit excitability.

To verify the role of elevated Kv7 function in decreased excitability of *Fmr1* KO MCs, we first confirmed that hypo-excitability is still observed in these cells when they were isolated from circuit activity using blockers of both glutamate and GABA receptors (as above). We found this to be the case, as evident by increased threshold, rheobase, and rheobase charge transfer (***Figure 3E–G***), as well as decreased number of APs (***Figure 3H***) in circuit-isolated MCs of *Fmr1* KO comparing to WT

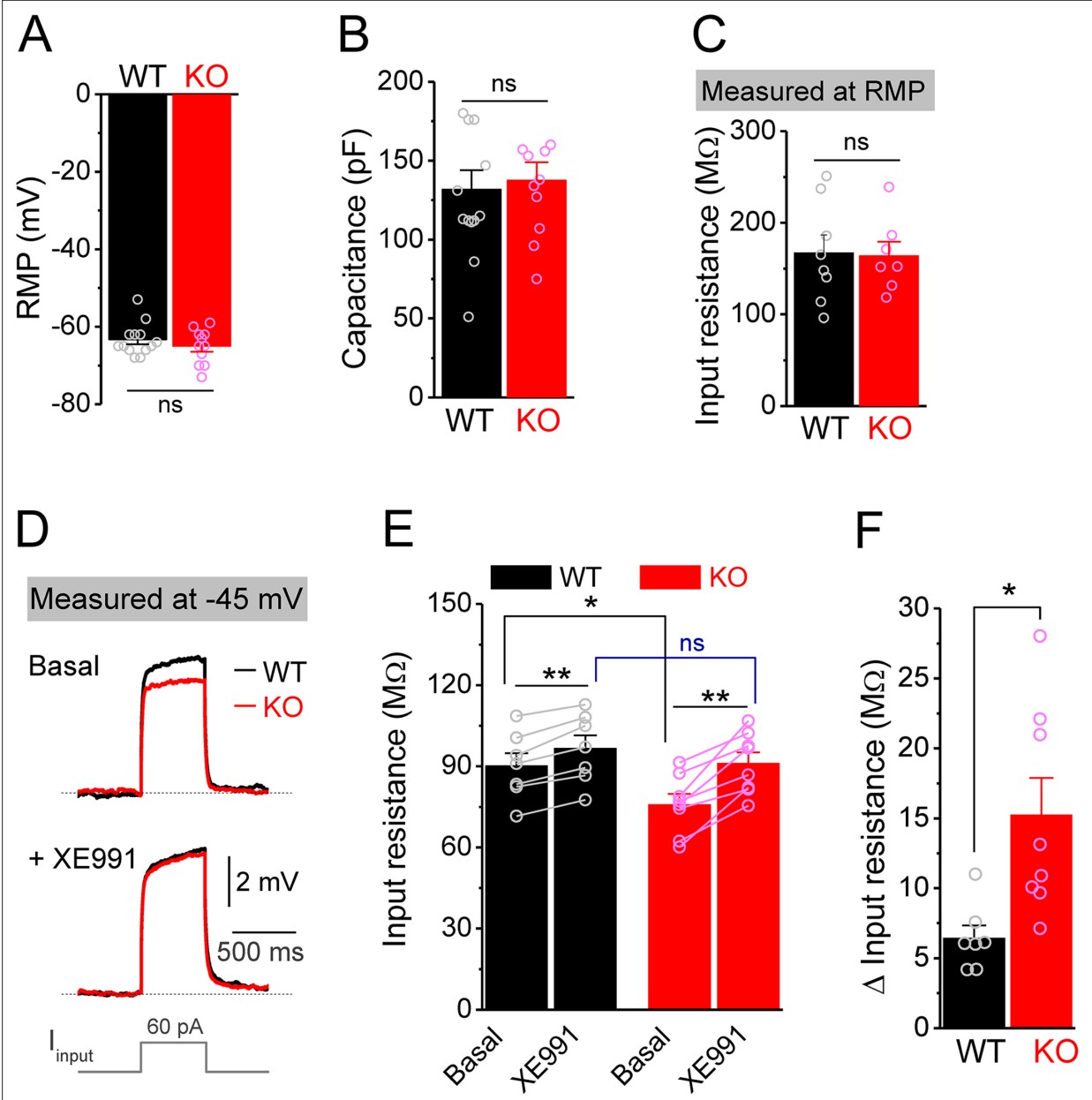

**Figure 2.** Decreased input resistance around threshold potential in *Fmr1* knockout (KO) mossy cells. (**A, B**) Resting membrane potential (RMP, **A**) and membrane capacitance (**B**) of mossy cells. (**C**) Input resistance measured at RMP level. (**D**) Input resistance measured at –45 mV. Sample traces of the depolarization current step (lowermost panel) induced voltage responses before (basal, upper panel) and during XE991 (+XE991, middle panel). (**E**) Summary data of input resistance before (basal) and during XE991. (**F**) Effects of XE991 on increasing of input resistance. Note XE991 have stronger effect on KO mossy cells. *p<0.05; **p<0.01; ns, not significant. The statistical data are listed in *Supplementary file 1*. Data are mean ± SEM.

mice. Importantly, Kv7 blocker XE991 abolished all these differences in excitability metrics between genotypes (*Figure 3E–H*).

Together, these results indicate that the hypo-excitability of MCs in *Fmr1* KO mice has primarily a cell-autonomous origin and can be attributed to abnormally elevated Kv7 function.

## MC hypo-excitability dominates adaptive circuit changes in *Fmr1* KO mice

Next we examined whether, in addition to cell-autonomous Kv7-mediated defects, there is also a circuit contribution to MC hypo-excitability. First, we measured spontaneous and miniature excitatory

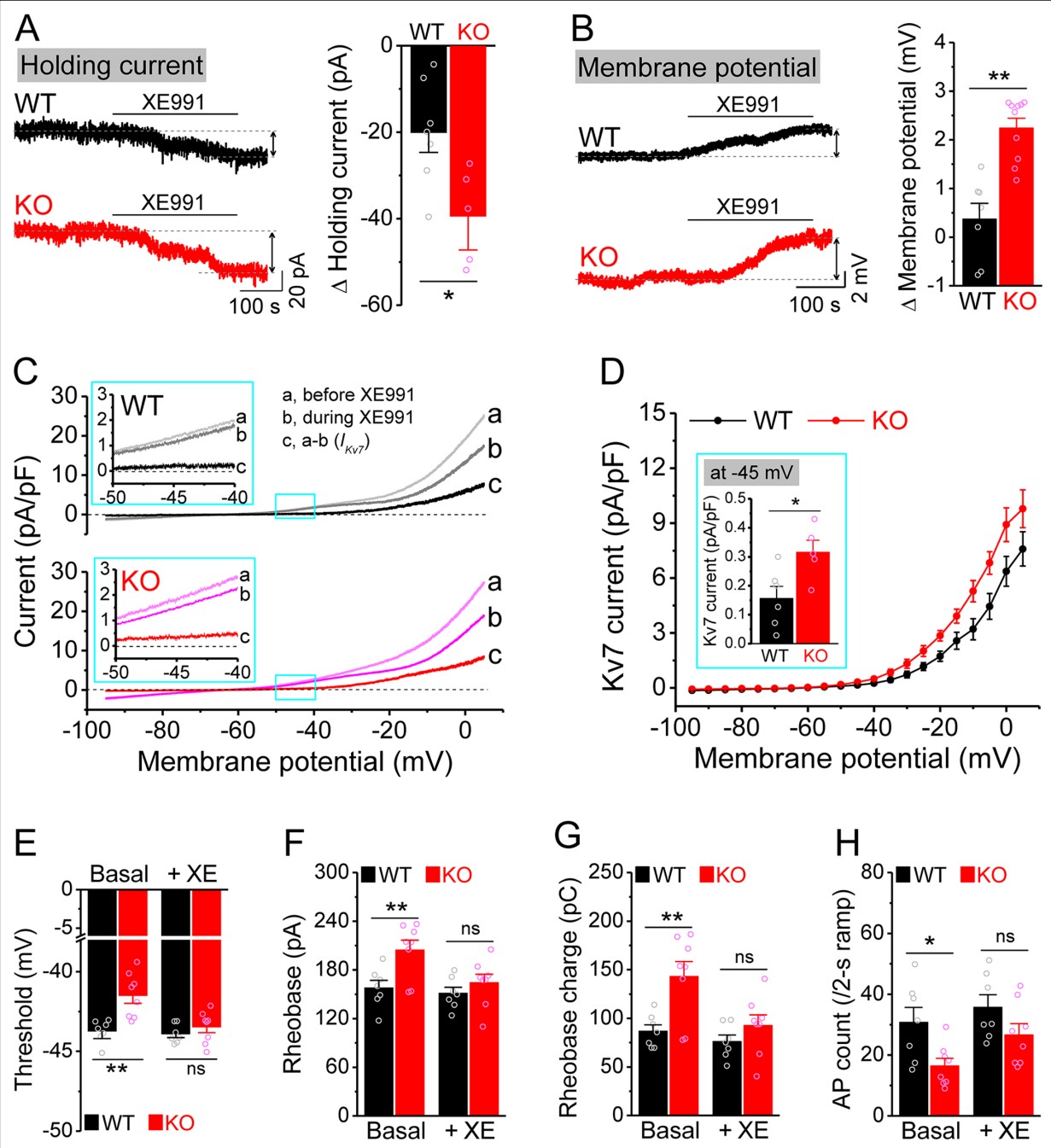

**Figure 3.** Enhanced Kv7 function causes hypo-excitability of *Fmr1* knockout (KO) mossy cells (MCs). (**A**) Changes in holding current at –45 mV in response to XE991. Left, sample traces; right, summary data. (**B**) Changes in membrane potential in response to XE991 when the initial potential being set at –45 mV. Left, sample traces; right, summary data. (**C**) Kv7 current was induced by a ramp protocol (from –95 to +5 mV with a rate of 0.02 mV/ms) and determined by XE991 sensitivity. Insets, the enlargements of boxed areas in main traces. (**D**) The I–V curves were constructed from ramp-evoked Kv7 currents every 5 mV (quasi-steady-state current, averages over 0.01 mV intervals) and normalized to respective cell capacitances. Inset, Kv7 current at –45 mV. (**E–H**) Increased threshold (**E**), rheobase (**F**), and rheobase charge transfer (**H**), as well as decreased number of action potentials (APs) (**H**) in pharmacologically isolated KO MCs. XE991 abolished these differences between genotypes. *p<0.05; **p<0.01; ns, not significant. The statistical data are listed in *Supplementary file 1*. Data are mean ± SEM.

The online version of this article includes the following source data and figure supplement(s) for figure 3:

**Figure supplement 1.** No changes in KCNQ2 and KCNQ3 expression in *Fmr1* knockout (KO) mice.

**Figure supplement 1—source data 1.** Original data for the western blot analysis in *Figure 3—figure supplement 1*.

**Figure supplement 1—source data 2.** Original data for the western blot analysis in *Figure 3—figure supplement 1* with relevant bands labeled.

postsynaptic currents (sEPSCs and mEPSCs) and found that the excitatory drive onto MCs was increased, as evident by the elevated number and instantaneous frequency of both sEPSCs and mEPSCs in KO MCs (*Figure 4A and B*, *Figure 4—figure supplement 1A and B*), without changes in their amplitudes (*Figure 4E*, *Figure 4—figure supplement 1C*). We note that this increase in excitatory drive cannot explain the MCs' hypo-excitability; rather these results likely reflect the hyperexcitable state of granule cells (*Deng et al., 2022*), which provide the majority of excitatory drive onto MCs. We next examined inhibitory inputs onto MCs and found that the number and instantaneous frequency of spontaneous inhibitory postsynaptic currents (sIPSCs) was also increased in KO MCs (*Figure 4C and D*) without changes in amplitude (*Figure 4F*). However, we did not observe significant differences in miniature inhibitory postsynaptic currents (mIPSCs) in KO MCs (*Figure 4—figure supplement 1D–F*). Because the overall circuit contribution to MC excitability is determined by the net effect of both excitatory and inhibitory drives, we examined the E/I ratio of these inputs, a parameter which integrates changes in both excitatory and inhibitory drives into a single variable and thus allows us to determine the net circuit effect on MCs excitability. Given that the amplitudes of sEPSCs and sIPSCs were indistinguishable between genotypes (*Figure 4E and F*), we could estimate the E/I ratio simply by using the mean frequency of spontaneous synaptic events (*Figure 4G and H*) and observed an increased E/I ratio in *Fmr1* KO MCs (↑~20%, *Figure 4I*) which acts in opposition to the intrinsic hypo-excitability observed in MCs. Thus, these adaptive circuit changes likely function as a compensatory mechanism to the intrinsic MCs' hypo-excitability in *Fmr1* KO mice, similarly to adaptive circuit changes also seen in the cortex (*Antoine et al., 2019*). However, this circuit adaptation is insufficient to compensate for MC excitability defects because we observed overall hypo-excitability in *Fmr1* KO MCs in the intact circuits.

## MC defects reduce excitatory drive onto hilar interneurons in *Fmr1* KO mice

The principal output of the dentate circuit is determined by granule cell firing, which is largely controlled by the balance of the excitatory inputs to granule cells from the stellate cells of the entorhinal cortex via the perforant path (PP), local excitatory inputs from MCs and inhibitory inputs from local interneurons. In the *Fmr1* KO mice, stellate cells have normal excitability (*Deng and Klyachko, 2016*); the granule cells are hyper-excitable (*Deng et al., 2022*), while MCs are hypo-excitable. We thus next probed excitability of the hilar interneurons, the remaining component of this circuit that has not been examined thus far. Because these inhibitory neurons exhibit both morphological and electrophysiological diversity and no clear correlation between morphology and electrophysiological properties has been observed (*Mott et al., 1997*), here we classified a total of 66 recorded interneurons into three types according to AP firing pattern: (1) fast-spiking interneurons (high-frequency non-adapting firing); (2) regular-spiking interneurons (slower and adapting firing); and (3) stuttering-like interneurons (high-frequency irregular bursting firing) (*Golomb et al., 2007*; *Figure 5—figure supplement 1A*). Chi-square test showed no significant differences in the ratios of the three types of interneurons between WT and KO mice (*Figure 5—figure supplement 1A*). Furthermore, we found that the passive membrane properties (RMP, capacitance and input resistance) and threshold of these interneurons were not significantly different between genotypes and among cell types (thus data were pooled, *Figure 5—figure supplement 1B–E*). These results indicate that the intrinsic excitability of hilar interneurons is not affected significantly in the absence of FMRP, suggesting that alterations in synaptic drive onto these cells may play a critical role in determining changes in local inhibition in *Fmr1* KOs.

Therefore, we next examined excitatory and inhibitory drives onto hilar interneurons by recording spontaneous and miniature synaptic inputs to these cells. We found that the number and instantaneous frequency of sEPSCs and mEPSCs were markedly decreased in *Fmr1* KO mice (*Figure 5A and B*, *Figure 5—figure supplement 1F and G*) with no changes in amplitudes (*Figure 5E*, *Figure 5—figure supplement 1H*). Further, the number and instantaneous frequency of sIPSCs were also decreased in KO interneurons (*Figure 5C and D*) without change in amplitude (*Figure 5F*), while no significant changes in mIPSCs was observed (*Figure 5—figure supplement 1I–K*). We then estimated the balance of E/I inputs onto interneurons using the same approach as above for MCs and observed a markedly decreased E/I ratio of inputs onto *Fmr1* KO interneurons (↓~60%, *Figure 5I*). We note that changes in the excitatory drive onto interneurons include both mEPSC and sEPSC frequencies, which

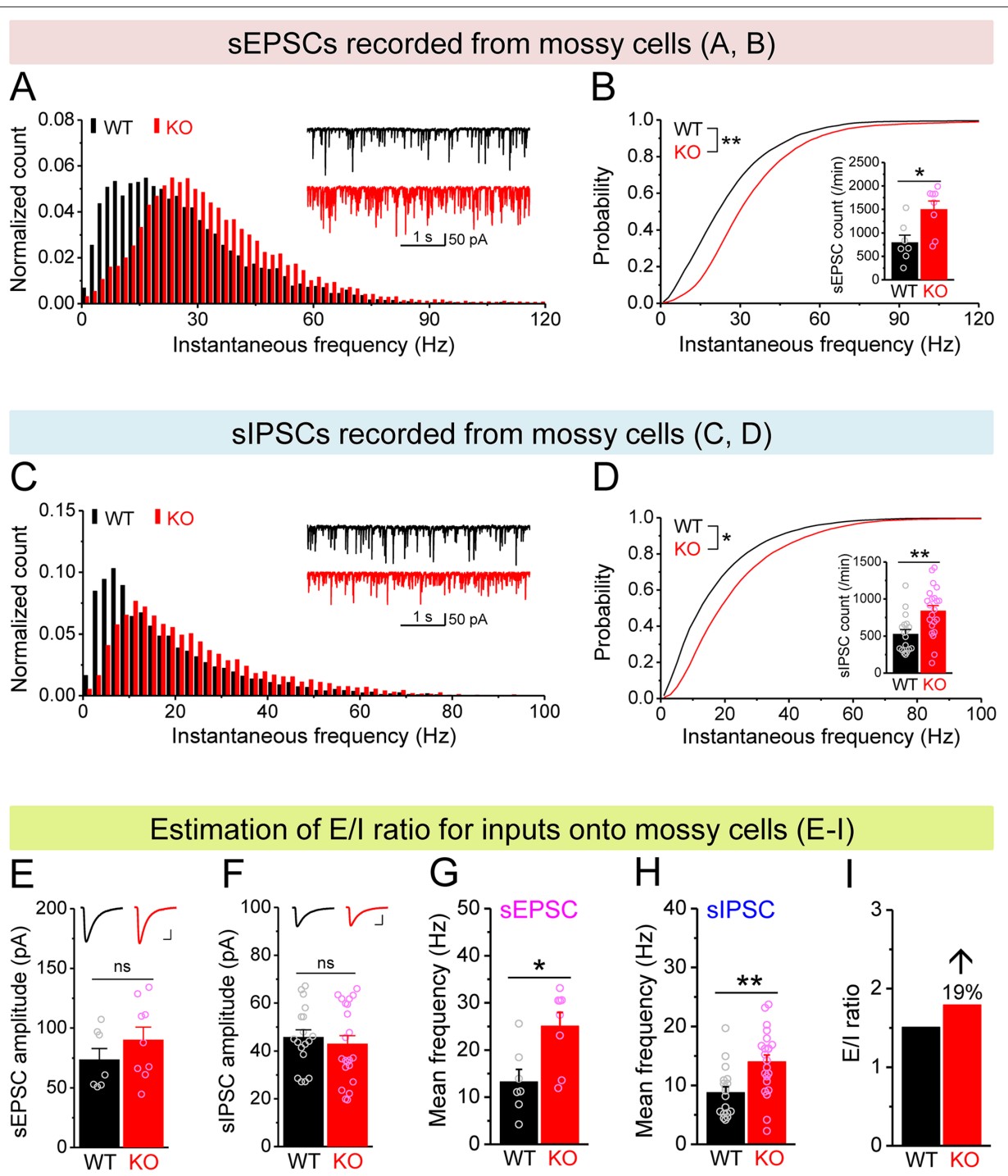

**Figure 4.** Increased excitation/inhibition (E/I) ratio of inputs onto *Fmr1* knockout (KO) mossy cells (MCs). (**A**) Distribution of spontaneous excitatory postsynaptic current (sEPSC) instantaneous frequency in MCs. A bin size of 2 Hz was used to calculate sEPSC frequency distribution from a 30-s-long trace per cell. The number of sEPSCs within each bin was normalized to the total number of the respective cells for pooling the data from all cells. Note that sEPSC events in KO mice had a shift toward high frequency. Inset, sample traces of sEPSCs for wildtype (WT) (black) and KO (red) mice. (**B**) Cumulative probability of sEPSC instantaneous frequency in MCs. Bar graph, number of sEPSCs per minutes. Note that both cumulative probability and number of sEPSCs reveal increased excitatory drive onto MCs. (**C, D**) Spontaneous inhibitory postsynaptic currents (sIPSCs) recorded from KO and WT MCs, aligned in the same way as in (**A, B**), respectively. The IPSC signals were downward here and also in *Figure 5*, *Figure 4—figure supplement 1*, and *Figure 5—figure supplement 1*, due to a high chloride electrode solution being used in these experiments. Note the increased sIPSC frequency and number in KO MCs. (**E, F**) Summary data for sEPSC amplitude (**E**) and sIPSC amplitude (**F**) recorded from MCs. Insets, sample sEPSC (**E**) and sIPSC (**F**) events for WT (black) and KO (red) MCs. Scale: 5 ms (horizontal) and 25 pA (vertical). (**G, H**) Mean frequency of sEPSCs (**G**) and sIPSCs (**H**) recorded

*Figure 4 continued on next page*

*Figure 4 continued*

from MCs. Note that loss of fragile X messenger ribonucleoprotein (FMRP) increased mean frequency of both sEPSC and sIPSC. (**I**) E/I ratio evaluated by sEPSC and sIPSC frequencies (mean values from **G** and **H**, respectively). Note the increased E/I ratio in *Fmr1* KO mice. *p<0.05; **p<0.01; ns, not significant. The statistical data are listed in **Supplementary file 1**. Data are mean ± SEM.

The online version of this article includes the following figure supplement(s) for figure 4:

**Figure supplement 1.** Changes in miniature synaptic inputs onto mossy cells (MCs) in *Fmr1* knockout (KO) mice.

reflect not only potential deficits in excitability of their input cells, such as MCs, but also changes in synaptic connectivity/function, that may arise from homeostatic circuit reorganization/compensation (see 'Discussion').

Considering that excitability of stellate cells is largely unaffected and granule cells show profound hyperexcitability in *Fmr1* KO mice, our observations suggest that it is the MCs that provide the dominant excitatory drive to hilar interneurons, causing a marked decrease in E/I ratio of inputs onto interneurons in *Fmr1* KO mice. To determine whether this is indeed the case, we took advantage of different properties of PP, granule cell, and MC axonal terminals (*Chancey et al., 2014*; *Chiu and Castillo, 2008*; *Shigemoto et al., 1997*). Specifically, granule cell and PP axonal terminals contain group II mGluRs (*Shigemoto et al., 1997*), while MC axonal terminals express type I cannabinoid (CB1) receptors (*Chancey et al., 2014*; *Chiu and Castillo, 2008*). Accordingly, mGluR group II agonist DCG-IV (1 µM) selectively inhibits granule cell- and PP-derived EPSCs onto hilar interneurons, while the CB1 agonist WIN 55212-2 (WIN, 5 µM) selectively inhibits MC-derived EPSCs onto these interneurons. In this analysis, for simplicity and better comparison among cells, we normalized the rate of sEPSCs to its own baseline before agonist application (i.e., normalized frequency). We observed that DCG-IV had little effect on the normalized frequency of sEPSCs recorded in interneurons in both genotypes (*Figure 6A, C, D and F*), indicating that granule cell- and PP-derived EPSCs comprise a limited portion of excitatory drive onto hilar interneurons in both genotypes. As a control for DCG-IV effectiveness, we observed that DCG-IV strongly reduced the normalized frequency of sEPSCs recorded from MCs that primarily receive excitatory inputs from granule cells (*Figure 6G–I*). In contrast, WIN markedly reduced the normalized frequency of sEPSCs in interneurons of both WT and KO mice (*Figure 6B, C, E and F*), while it failed to affect the normalized frequency of sEPSC recorded from MCs (*Figure 6H and I*), suggesting that MCs provide a significant proportion of excitatory drive onto hilar interneurons. Together with the observations above, this analysis supports the notion that the hypo-excitability of MCs in *Fmr1* KO mice is a major factor contributing to the reduction of excitatory drive onto hilar interneurons, which ultimately results in reduced local inhibition.

## Circuit-wide inhibition of Kv7 channels increased local inhibitory drive and abolished differences in granule cell excitability between genotypes

If upregulation of Kv7 function is largely limited to MCs in the dentate circuit of *Fmr1* KO mice, then our findings suggest that inhibition of Kv7 channels would be more efficient to enhance MC excitability in KO than WT mice and thus may be more effective to boost the inhibitory drive onto *Fmr1* KO granule cells, thus reducing granule cell hyperexcitability in KO mice. To test this hypothesis, we first simultaneously recorded sEPSCs and sIPSCs in the granule cells to get a better quantitative analysis of the E/I input onto these cells. Granule cells were held at –40 mV, resulting in sEPSC appearing as a downward current, with sIPSC appearing as upward current, which was verified by selective blockers of GABA$_A$ or AMPA and NMDA receptors (*Figure 7—figure supplement 1A*). We found that loss of FMRP did not affect the mean frequencies, amplitudes, and charge transfer of sEPSCs and sIPSCs onto granule cells (*Figure 7B and C*, *Figure 7—figure supplement 1B–G*), and thus did not alter the baseline E/I ratio as determined by any of these parameters (*Figure 7D*, *Figure 7—figure supplement 1D–G*). In line with our hypothesis, circuit-wide inhibition of Kv7 channels with bath application of XE991 markedly increased sIPSC frequency in the granule cells of *Fmr1* KO but not in WT mice (*Figure 7C*), while having no effect on sEPSC frequency in either genotype (*Figure 7B*). As a result, XE991 application shifted the E/I balance toward stronger inhibition in the *Fmr1* KO, but not in WT mice (*Figure 7D*). These results indicate that circuit-wide inhibition of Kv7 channels could be effective in reducing granule cell hyperexcitability in the KO mice. We noted that XE991 also slightly

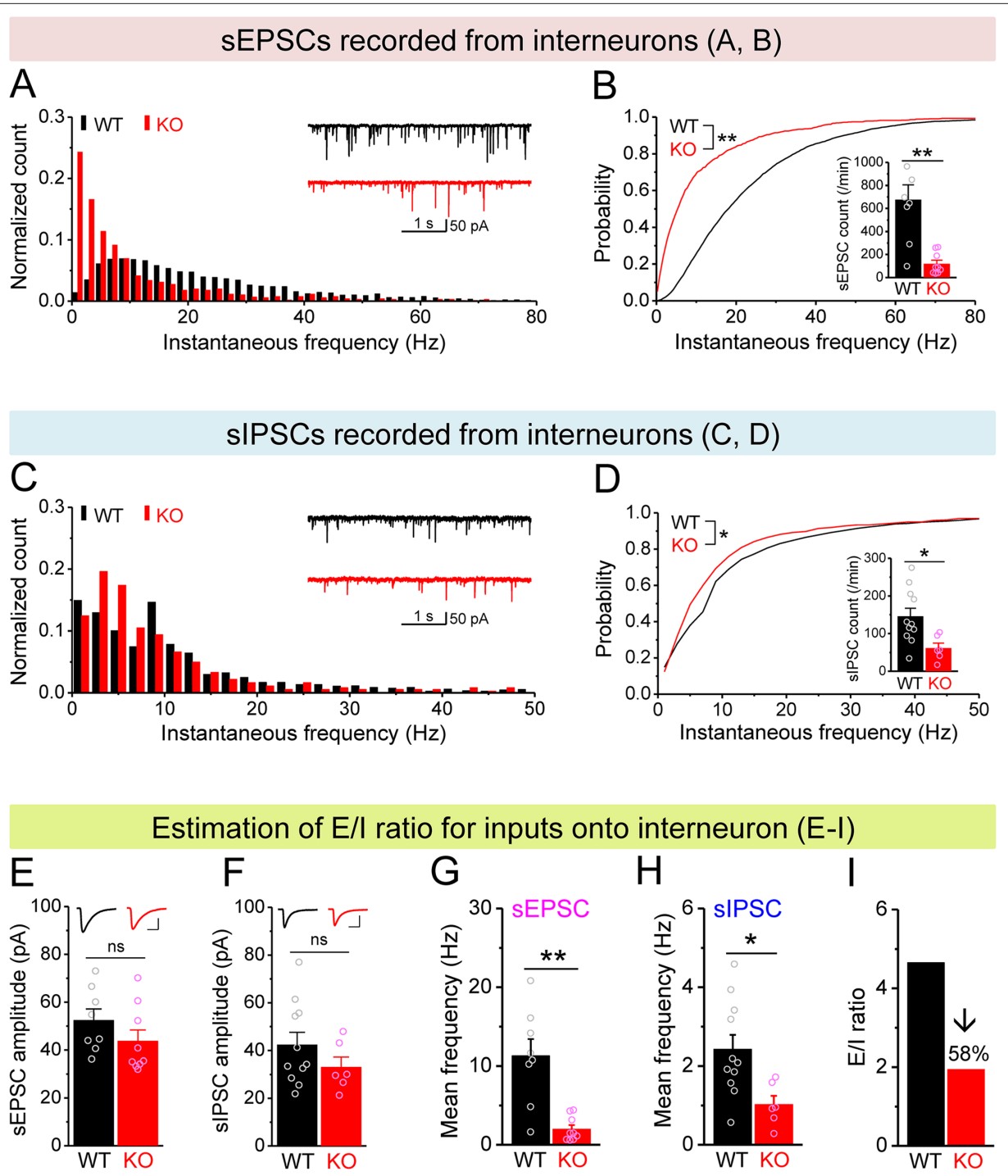

**Figure 5.** Decreased excitation/inhibition (E/I) ratio of inputs onto *Fmr1* knockout (KO) hilar interneurons. (**A**) Distribution of spontaneous excitatory postsynaptic current (sEPSC) instantaneous frequency in hilar interneurons. Note that KO sEPSC events had a shift toward low frequency. Inset, sample traces of sEPSCs for wildtype (WT) (black) and KO (red) interneurons. (**B**) Cumulative probability of sEPSC instantaneous frequency in interneurons. Bar graph shows number of sEPSCs per minutes. Note that both cumulative probability and number of sEPSCs reveal decreased excitatory drive onto interneurons. (**C, D**) Spontaneous inhibitory postsynaptic currents (sIPSCs) recorded from interneurons, aligned in the same way as in (**A, B**), respectively. Note the decreased sIPSC frequency and number in KO interneurons. (**E, F**) Summary data for sEPSC amplitude (**E**) and sIPSC amplitude (**F**) recorded from interneurons. Insets, sample sEPSC (**E**) and sIPSC (**F**) events for WT (black) and KO (red) mossy cells (MCs). Scale: 5 ms (horizontal) and 25 pA (vertical). (**G, H**) Mean frequency of sEPSCs (**G**) and sIPSCs (**H**) recorded from interneurons. Note loss of FMRP decreased mean frequency of both sEPSC and sIPSC. (**I**) E/I ratio evaluated by sEPSC and sIPSC frequencies in interneurons (mean values from **G** and **H**, respectively). Note the decreased E/I ratio in *Fmr1* KO mice. *p<0.05; **p<0.01; ns, not significant. The statistical data are listed in ***Supplementary file 1***. Data are mean ± SEM.

*Figure 5 continued on next page*

*Figure 5 continued*

The online version of this article includes the following figure supplement(s) for figure 5:

**Figure supplement 1.** Changes in miniature synaptic inputs onto hilar interneurons in *Fmr1* knockout (KO) mice.

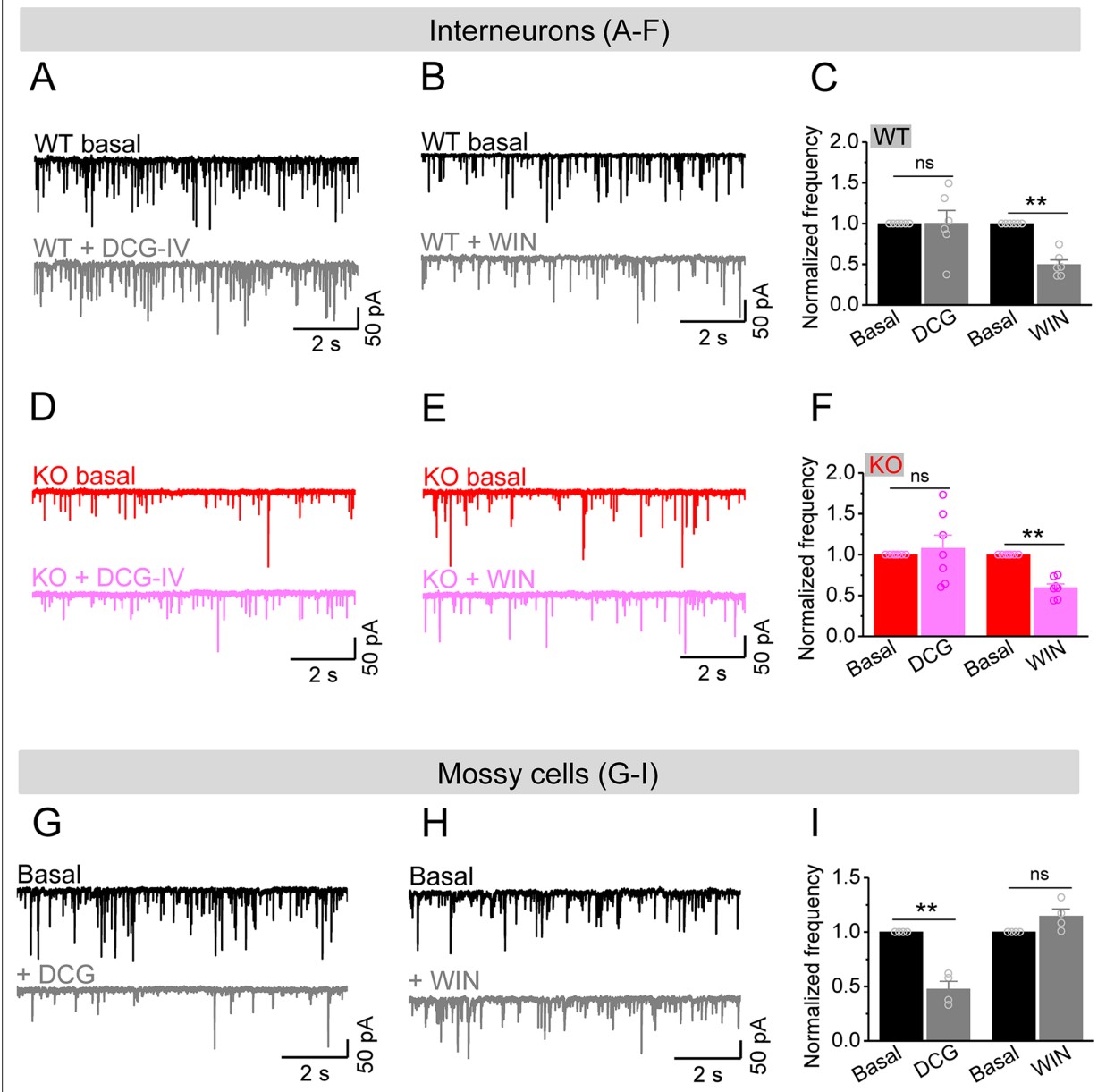

**Figure 6.** Mossy cells (MCs) provide the main excitatory drive onto hilar interneurons. (**A**) Sample traces of spontaneous excitatory postsynaptic currents (sEPSCs) recorded from an interneuron of wildtype (WT) mouse before and during DCG-IV. (**B**) The same as in (**A**), but for WIN55212-2 in an interneuron of WT mouse. (**C**) Effect of DCG-IV or WIN55212-2 on the sEPSC normalized frequency recorded from interneurons of WT mice. (**D–F**) The same as in (**A–C**) but for interneurons of knockout (KO) mice. Note that WIN55212-2 had comparable effects on normalized frequency of sEPSCs in KO and WT interneurons, but DCG-IV did not have measurable effects on both genotypes. (**G**) Control experiment showing effectiveness of DCG-IV. Sample traces of sEPSCs recorded from MCs before and during DCG-IV. (**H**) Sample traces of sEPSCs recorded from MCs before and during WIN 55212-2. (**I**) Summary data of changes in the normalized frequency of MC sEPSCs in response to DCG-IV (47.6 ± 7.2% of basal) and WIN55212-2 (114.5 ± 6.7% of basal). Note that, compared to interneurons (**A–F**), MCs exhibited opposite response to these two agonists, indicating the effectiveness of both agonists. **p<0.01; ns, not significant. The statistical data are listed in *Supplementary file 1*. Data are mean ± SEM.

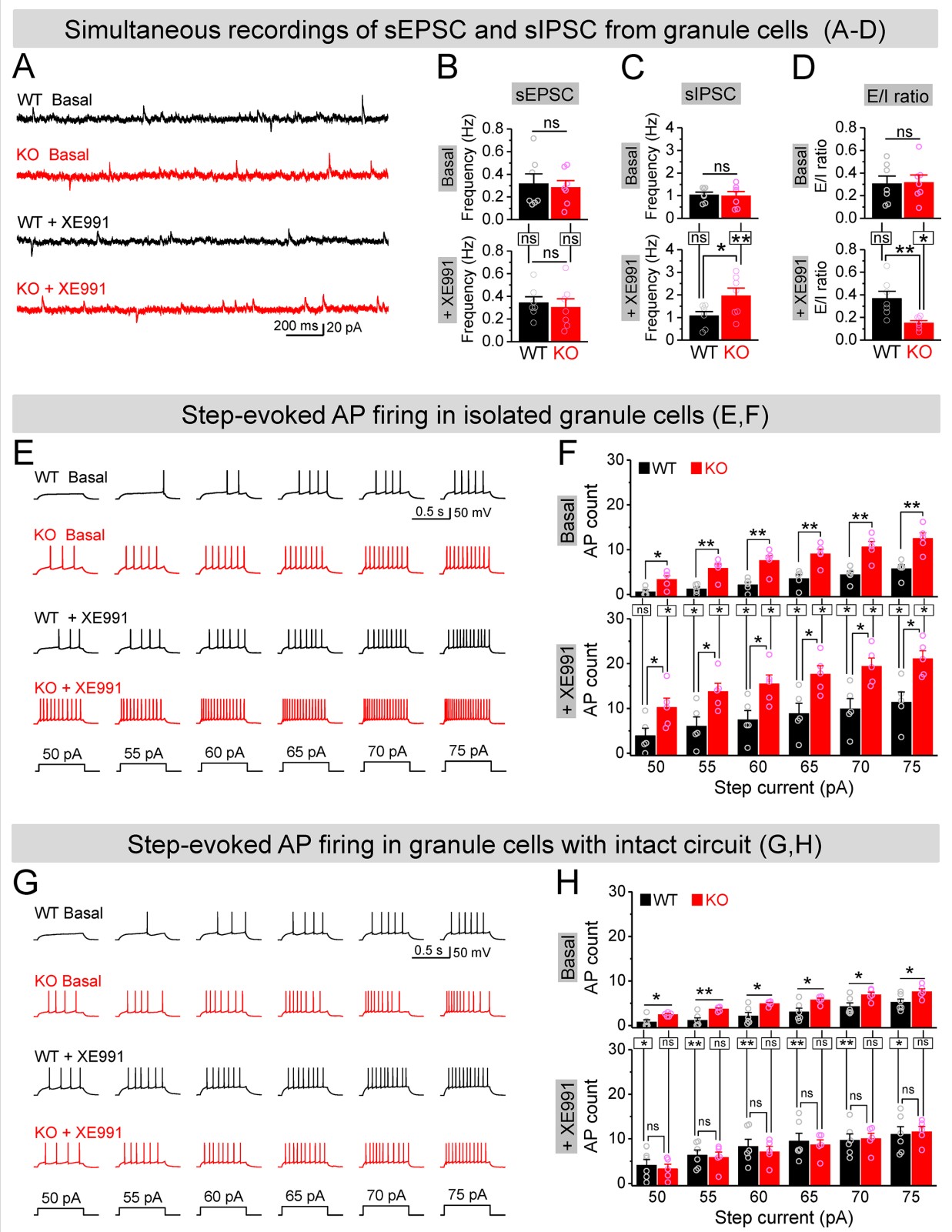

**Figure 7.** Circuit-wide inhibition of Kv7 channels boosted inhibitory drive onto granule cells in *Fmr1* knockout (KO) mice. (**A**) Sample traces of simultaneous recording of spontaneous excitatory postsynaptic current (sEPSC) (downward events) and spontaneous inhibitory postsynaptic current (sIPSC) (upward events) from granule cells (also see *Figure 7—figure supplement 2A*). (**B**) Summary data for sEPSC mean frequency in basal (upper) and during XE991 (lower). Horizontal lines (with or without dropdown) denote comparison between genotypes; vertical lines indicate comparison

*Figure 7 continued on next page*

*Figure 7 continued*

between before and during XE991 within genotypes. (**C**) The same as in (**B**), but for sIPSC simultaneously recorded from the same granule cells. Note that XE991 increase sIPSC frequency only in KO mice, but not in wildtype (WT) mice. (**D**) Excitation/inhibition (E/I) ratio evaluated by frequency. XE991 (lower) significantly decreased the E/I ratio in KO mice only. (**E**) Evaluation of granule cell excitability by recording action potentials (APs). Sample traces for multistep-current (lowermost panel) evoked APs in WT and KO granule cells in the pharmacologically isolated granule cells in the absence (upper) or presence of XE991 (lower). (**F**) Summary data for number of APs exemplified in (**E**) showing increased excitability of the pharmacologically isolated granule cells in the absence (upper) or presence of XE991 (lower). Also, note that XE991 increased number of AP in both WT and KO granule cells. Horizontal lines (with dropdown) denote comparison between genotypes; vertical lines indicate comparison between in the absence and presence of XE991 within genotypes. (**G**) The same as in (**E**), but for granule cells with intact dentate circuit. (**H**) The same as in (**F**), but for granule cells with intact circuit. Note that XE991 abolished the difference of AP count between genotypes. *p<0.05; **p<0.01; ns, not significant. The statistical data are listed in *Supplementary file 1*. Data are mean ± SEM.

The online version of this article includes the following figure supplement(s) for figure 7:

**Figure supplement 1.** Effect of XE991 on spontaneous synaptic inputs onto granule cells.

**Figure supplement 2.** Estimation of direct and circuit effects of XE991 on granule cell (GC) excitability.

reduced both sEPSC and sIPSC amplitudes, but to a similar extent in both genotypes (*Figure 7—figure supplement 1B and C*), and without detectable changes in charge transfer (*Figure 7—figure supplement 1E and F*).

To examine more directly whether circuit-wide inhibition of Kv7 channels could be effective in reducing granule cell hyperexcitability in the KO mice, we performed analyses of granule cell firing – the ultimate measure of cellular excitability. APs were evoked by a multistep current injection (55–75 pA, *Figure 7E*). To disentangle the direct- and circuit-mediated actions of Kv7 inhibition, we first synaptically isolated granule cells from the circuit using the same pharmacological approach as described above and observed that loss of FMRP strongly increased the AP firing in granule cells (*Figure 7E and F*, upper panels), as we reported previously (*Deng et al., 2022*). Inhibition of Kv7 channels with XE991 (10 μM) further increased AP firing in both genotypes, with significantly higher number of APs still observed in the isolated granule cells of *Fmr1* KO mice (*Figure 7E and F*, lower panels). These results indicate that intrinsic hyperexcitability of granule cells in *Fmr1* KO mice is largely unrelated to Kv7 channels. Indeed, we have demonstrated that this intrinsic defect is caused by abnormal extrasynaptic GABA$_A$ receptor activity in the absence of FMRP (*Deng et al., 2022*). We then performed the same recordings in granule cells with intact dentate circuit and again observed markedly increased excitability in the *Fmr1* KO, as evident by much larger number of APs fired by KO than WT granule cells (*Figure 7G and H*, upper panels). In line with our hypothesis, circuit-wide inhibition of Kv7 channels with XE991 abolished the difference in the number of APs between genotypes (*Figure 7G and H*, lower panels).

To further clarify this notion, we compared the effects of XE991 between isolated- (*Figure 7F*) and circuit-intact granule cells (*Figure 7H*), and probed the origin of XE991 contribution to granule cell excitability. Specifically, even though the direct action of XE991 on granule cells was to increase excitability in both genotypes (*Figure 7—figure supplement 2A*), the circuit action of XE991 resulted in suppressing granule cell excitability in *Fmr1* KO mice (↓~6 APs, *Figure 7—figure supplement 2B*), while having minimal effect in WT mice (changes fluctuating around 0, *Figure 7—figure supplement 2B*). This is in line with the observations above that XE991 increased sIPSC frequency but not sEPSC frequency, and only in the KO but not in WT mice (*Figure 7A–D*). Together with our findings that the interneuron activity is driven in a large part by excitatory input from the MCs, these results indicate that the primary site of action of XE991 is the MCs in *Fmr1* KO mice. Thus, our findings support the notion that circuit-wide inhibition of Kv7 channels boosted up inhibitory drive onto granule cells and abolished differences in granule cell excitability between genotypes by enhancing the local inhibitory circuit function.

## Implications of MC hypo-excitability and circuit E/I imbalance to dentate function

Granule cell excitability and dentate output are not only dependent on basal synaptic inputs (i.e., basal E/I balance, as evident in *Figure 7*), but more importantly these cellular/circuit functions are controlled by a delicately tuned dynamic E/I balance during circuit activity. By providing the major excitatory drive to local interneurons, MCs play a key role in controlling circuit inhibition onto granule

cells via a three-synapse pathway (granule cells→MCs→interneurons→granule cells, *Figure 8—figure supplement 3A*). Accordingly, we hypothesized that the Kv7-dependent hypo-excitability of MCs in the *Fmr1* KO mice compromises the effectiveness of this critical inhibitory pathway.

To test this possibility, we recorded compound postsynaptic current (cPSC) in granule cells by stimulating PP and holding cells at −45 mV. This holding potential was chosen as it was an intermediate potential between the excitatory and inhibitory reversal potentials ensuring comparable driving force for excitatory and inhibitory conductances. The PP-stimulation-evoked cPSC is the summation of largely overlapping excitatory and inhibitory postsynaptic currents, which shows as an initial downward excitatory component followed by an upward inhibitory component (*Figure 8—figure supplement 1A*). At the end of each recording, the pure EPSC was isolated by adding GABA$_A$ receptor blocker gabazine (5 µM) (*Figure 8—figure supplement 1B*), which then was averaged to create an EPSC template for each cell in order to isolate underlying EPSC and IPSC from the cPSC and calculate E/I ratio (*Figure 8—figure supplement 1C and D*). For better comparison among cells, we normalized the cPSC and underlying IPSC to their respective underlying EPSC, which reflects the PP stimulation intensity (*Figure 8—figure supplement 1E*).

In line with our observations above, loss of FMRP significantly decreased the inhibitory component of cPSC and the underlying IPSC (*Figure 8D and E*, upper panels), while the excitatory component of cPSC was comparable between genotypes (*Figure 8C*, upper panel). As a result, E/I ratio of the inputs onto granule cells was abnormally increased in *Fmr1* KO mice (*Figure 8F and G*, upper panels). Moreover, we also observed a wider excitation window (defined as the duration of cPSC excitatory component, *Figure 8—figure supplement 1C*) in the granule cells of the KO mice (*Figure 8H*, upper panel). In support of our hypothesis, circuit-wide inhibition of Kv7 with XE991 significantly increased the inhibitory component of cPSC and underlying IPSC in *Fmr1* KO (*Figure 8D and E*, lower panels), but not in WT mice. Consequently, inhibition of Kv7 normalized E/I ratio (*Figure 8F and G*, lower panels), as well as the excitation window in the granule cells of *Fmr1* KO mice (*Figure 8H*, lower panel). These results indicate that Kv7-dependent hypo-excitability of MCs in *Fmr1* KO mice is a critical defect in the dentate circuit and that circuit-wide inhibition of Kv7 is sufficient to normalize circuit E/I balance.

Finally, we asked what are the functional consequences of MC hypo-excitability and dentate circuit E/I imbalance in *Fmr1* KO mice? The power of theta–gamma oscillations is particularly high in the dentate gyrus (*Csicsvari et al., 2003*) and plays a critical role in many dentate functions, such as pattern separation (*Leutgeb et al., 2007*), information coding (*Mizuseki et al., 2009*; *Pernía-Andrade and Jonas, 2014*), and thus learning/memory (*Bott et al., 2016*; *Lisman and Jensen, 2013*; *Neves et al., 2022*). In order to evaluate the contribution of MC defects in *Fmr1* KO mice to this critical dentate function, we used a protocol of double-oscillation interplay at the single-cell level (*Hasselmo et al., 2007*; *Mircheva et al., 2019*). In this paradigm, gamma (~50 Hz) frequency stimulation of PP suppresses granule cell output driven by PP stimulation at the theta (~5 Hz) frequency range (gamma suppression, for brevity). The experiments included two stimulation protocols: a control protocol at 5 Hz (theta stimulation, whose intensity was adjusted to achieve AP probability of ~0.5 and the intensity was then kept in the following test protocol for the same cell); and a test protocol with a gamma burst (five stimuli at 50 Hz) included 200 ms before the 5 Hz theta train. Theta stimulation of the PP evoked a steady baseline firing in granule cells (*Figure 8I–K*, upper panels). In line with previous studies (*Hasselmo et al., 2007*; *Mircheva et al., 2019*), a preceding gamma stimulation suppressed granule cell output in response to theta train in both genotypes (*Figure 8I–K*, upper panels). However, this gamma suppression was much less efficient in the KO than in WT mice, as evident by the significant higher AP probability in KO mice (*Figure 8I–K*, upper panels). Considering that granule cells are the only output neurons in the dentate gyrus and thus integrate all dentate circuit operations, these results indicate that loss of FMRP causes abnormal dentate information processing, leading to excessive dentate output.

The dentate GABAergic system plays a critical role in the above gamma suppression of granule cell output (*Mircheva et al., 2019*). Based on our findings, we hypothesized that the compromised dentate inhibitory pathway in the *Fmr1* KO mice weakens gamma suppression by enhancing the EPSP summation in granule cells. The EPSP measured in granule cells in response to the PP stimulation integrates both excitatory and inhibitory synaptic inputs onto granule cells, including the direct synaptic input from the PP and all the PP stimulation-associated feedforward and feedback synaptic inputs. In other words, the EPSP in granule cells integrates all dentate circuit 'operations'. Mechanistically, once

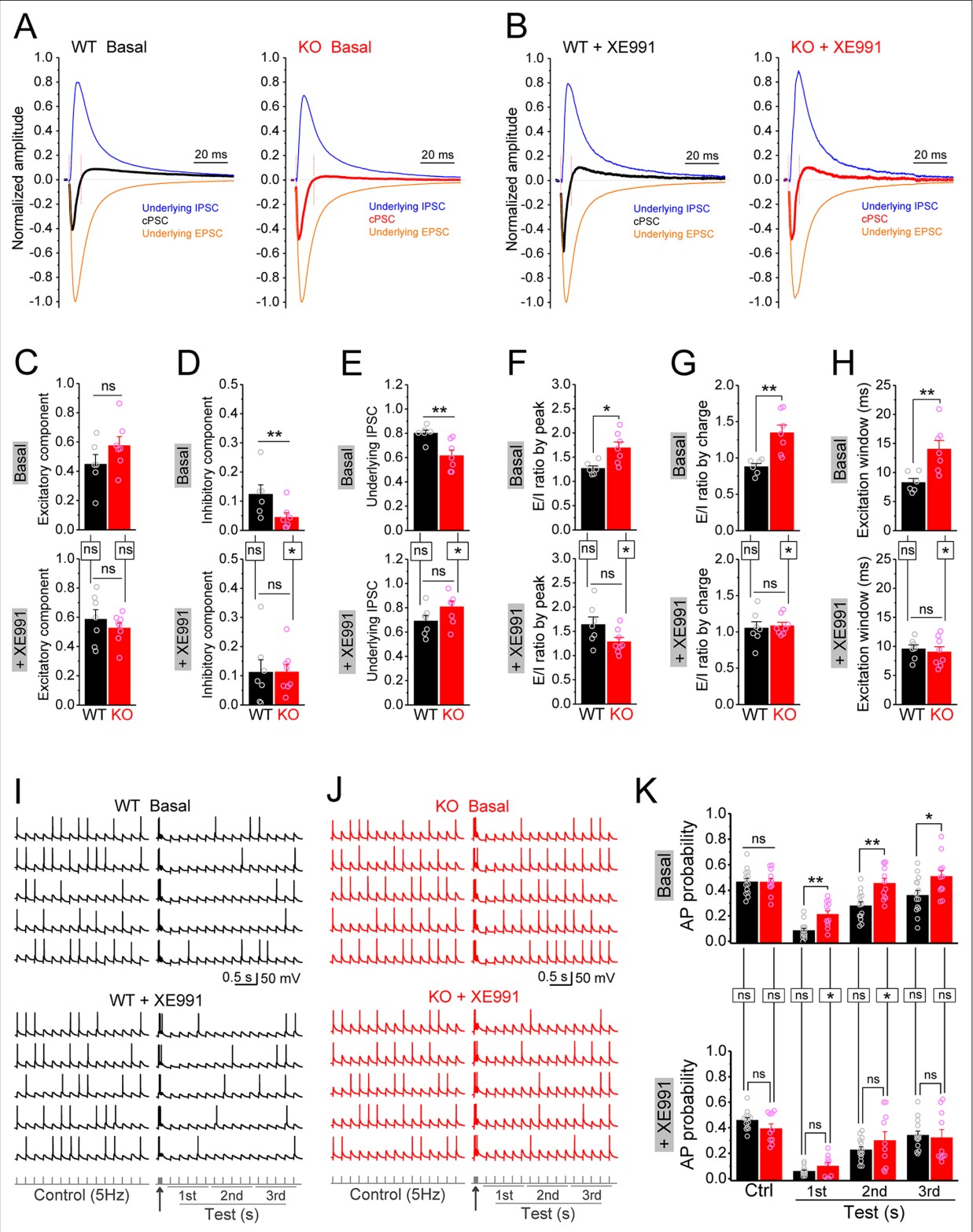

**Figure 8.** Circuit-wide inhibition of Kv7 channels restored dentate output during theta–gamma coupling stimulation in *Fmr1* knockout (KO) mice. (**A**) Sample traces of perforant path (PP)-stimulation-evoked compound postsynaptic currents (cPSC) and their respective underlying excitatory postsynaptic current (EPSC) and inhibitory postsynaptic current (IPSC), in the basal state. For better comparison, the traces were normalized to their own underlying EPSC, which reflects stimulation intensity. Red vertical lines denote the excitation window that is summarized in (**H**). Stimulation artifacts were removed and baseline before stimulation was shifted to be 0 for presentation purpose. (**B**) The same as in (**A**), but in the presence of XE991.

*Figure 8 continued on next page*

*Figure 8 continued*

(**C–E**) Summary data of normalized excitatory component (**C**), inhibitory component (**D**), and underlying IPSC (**E**), in the absence (basal, upper panels) and presence of XE991 (+XE991, lower panels). (**F**) Summary data of excitation/inhibition (E/I) ratio evaluated by the peaks of underlying EPSC and IPSC in the absence (basal, upper panel) and presence of XE991 (+XE991, lower panel). (**G**) The same as in (**F**), but evaluated by the charge transfers of underlying EPSC and IPSC. (**H**) Summary data of excitation window in the absence (basal, upper panel) and presence of XE991 (+XE991, lower panel). (**I**) Sample traces of theta–gamma coupling stimulation-evoked action potentials (APs) in granule cells from wildtype (WT) mice, in the absence of (basal, upper panel) or presence of XE991 (+XE991, middle panel). Lower panel shows stimulation protocols: control, 15 stimuli at 5 Hz; test, a burst of gamma stimulation (five stimuli at 50 Hz, arrow) 200 ms before 15 stimuli at 5 Hz. AP probability in test train was calculated in 1-s-bin (i.e., binned in first, second, or third second) and plotted in (**K**). (**J**) The same as in (**I**), but for KO mice. (**K**) Summary data of gamma suppression of dentate output in response to PP stimulation at theta frequency in the absence of (basal, upper) or presence of XE991 (+XE991, lower). Note loss of fragile X messenger ribonucleoprotein (FMRP) compromised gamma suppression of AP output in granule cells, and Kv7 blocker XE991 restored the gamma burst-induced suppressive effect on dentate output in *Fmr1* KO mice. *p<0.05; **p<0.01; ns, not significant. Horizontal lines (with or without dropdown) denote comparison between genotypes; vertical lines indicate comparison between in the absence and presence of XE991 within genotypes. The statistical data are listed in ***Supplementary file 1***. Data are mean ± SEM.

The online version of this article includes the following figure supplement(s) for figure 8:

**Figure supplement 1.** Isolation of underlying excitatory postsynaptic current (EPSC) and inhibitory postsynaptic current (IPSC) from compound postsynaptic current (cPSC).

**Figure supplement 2.** Circuit-wide inhibition of Kv7 channels enhanced the gamma burst-induced suppression of EPSP (excitatory postsynaptic potential) integration in *Fmr1* knockout (KO) mice.

**Figure supplement 3.** Circuit-wide inhibition of Kv7 channels corrected dentate gyrus output in *Fmr1* knockout (KO) mice.

the integrated EPSP reaches the threshold level, it triggers an AP, which makes the EPSP amplitude a direct predictor of AP firing. We found that theta stimulation evoked a steady EPSP (***Figure 8— figure supplement 2A and B***), and a preceding gamma burst markedly suppressed the EPSP in WT mice (***Figure 8—figure supplement 2A and B***). In line with our results above, this suppression effect was significantly weaker in KO mice (***Figure 8—figure supplement 2A and B***), suggesting that the compromised dentate inhibitory pathway weakens gamma suppression by enhancing the EPSP summation in the KO mice (***Figure 8—figure supplement 3B***). If this is the case, our results predict that circuit-wide inhibition of Kv7 channels should dampen EPSP summation and restore the gamma suppression. Indeed, we found that Kv7 blocker XE991 reduced the EPSP amplitude in KO mice to the WT level (***Figure 8—figure supplement 2C and D***) and, most importantly, normalized gamma suppression of granule cell output during theta activity in *Fmr1* KO mice (***Figure 8J–K***, ***Figure 8— figure supplement 3C***).

Collectively, these findings demonstrate that MCs' hypo-excitability in *Fmr1* KO mice results in circuit E/I imbalance that compromises the dentate information processing functions, and that circuit-based interventions, such as inhibition of Kv7, could be a potential therapeutic strategy to re-normalize the circuit E/I balance and ameliorate dentate dysfunction in FXS.

## Discussion

Dentate gyrus, the information gateway to the hippocampus, receives and processes the vast majority of cortical-hippocampal inputs. It carries out numerous information-processing tasks to form unique memories and conveys the information to the CA3 region. The precisely controlled E/I balance in the dentate circuit is critical to many fundamental computations that are believed to underlie learning and memory (***Amaral et al., 2007***). The dentate gyrus output is controlled by the local networks whose activity is determined not only by the entorhinal cortical inputs and local interneuron activity, but also by another type of local excitatory neurons, the MCs (***Scharfman, 2016***; ***Scharfman, 2018***). These cells have long been considered 'enigmatic' due to their complex and incompletely understood roles in the circuit, yet recently the critical roles of MCs in dentate computations and spatial memory have emerged (***Amaral et al., 2007***; ***Berron et al., 2016***; ***Botterill et al., 2021b***; ***Fredes and Shigemoto, 2021***; ***Scharfman, 2016***). Moreover, MC defects were also found to play major roles in epilepsy and many neurodevelopmental disorders (***Pelkey et al., 2017***; ***Scharfman, 2016***; ***Scharfman, 2018***; ***Noebels et al., 2012***). Whether MC defects are present and contribute to hippocampal dysfunction in FXS has remained largely unknown. Here, we demonstrate that loss of FMRP causes profound MC hypo-excitability, an unusual defect which is opposite to cellular hyperexcitability reported in all other

excitatory neurons in the FXS mouse model, including dentate granule cells (**Deng et al., 2022**). Since MC hypo-excitability was observed both in the synaptically isolated MCs and with the intact circuit, this defect has a cell-autonomous origin and the adaptive circuit changes were insufficient to compensate for it.

In pursuing the mechanism of this defect, we discovered that it is caused by abnormally elevated Kv7 function in the absence of FMRP. While Kv7's contribution to input resistance is negligible at the resting potential of MCs (below –60 mV) (**Delmas and Brown, 2005**; **Jones et al., 2021**; **Liu et al., 2021**), once the membrane potential rises to a near-threshold level (–45 mV), the excessive activity of Kv7 channels acts to reduce input resistance and suppresses AP initiation, thus reducing MC excitability in *Fmr1* KO mice. How does loss of FMRP cause an increase in Kv7 function? Both KCNQ2 and KCNQ3 have been identified as FMRP translational targets (**Darnell et al., 2011**). Our western blot analysis showed that the expression levels of KCNQ2 and KCNQ3 in KO mice were not altered in the whole brain lysate in general, nor in the dentate gyrus region, specifically. In addition to translational regulation, FMRP is also known to regulate activity of several K$^+$ channels, including Slack, BK, SK, and Kv1.2 via protein–protein interactions (**Deng and Klyachko, 2021**). However, in all of these cases, FMRP loss decreases rather than increases K$^+$ channel activity, making loss of protein–protein interactions an unlikely mechanism to explain increased Kv7 function. One alternative possibility is that loss of FMRP affects intracellular second-messenger signaling pathways that may regulate Kv7 activity (**Haick et al., 2017**). For example, activation of PKC suppresses Kv7 function (**Delmas and Brown, 2005**; **Haick et al., 2017**) and PKCε expression was found to be reduced in hippocampus of *Fmr1* KO mice (**Marsillo et al., 2021**), supporting a possibility that the increased Kv7 function in KO MCs may be associated with abnormal PKC signaling. However, further studies will be needed to elucidate the precise mechanism responsible for the increased Kv7 function in *Fmr1* KO mice.

Our analyses show that while the MC excitability dysfunction is unusual in being opposite in direction to other excitatory neurons, it nevertheless leads to abnormally increased granule cell output (i.e., dentate output) in *Fmr1* KO mice. This counterintuitive effect arises because in addition to directly exciting granule cells, MCs act via local inhibitory networks to indirectly suppress granule cell activity. Therefore, the net effect of MC activity on granule cell output is orchestrated via a dynamically tuned E/I balance of direct excitation and indirect inhibition. In their control of inhibition, the MCs relay and expand granule cell-derived excitatory drive onto interneurons, which then convert this drive into stronger feedback inhibition onto granule cells (**Botterill et al., 2021a**; **Hashimotodani et al., 2017**; **Houser et al., 2021**; **Yeh et al., 2018**). Notably, the resulting E/I balance driving granule cell output is not static; rather, during network activity, short-term plasticity actively modulates outputs from both excitatory and inhibitory synapses to dynamically fine-tune the E/I drives and optimize neural computations (**Bartley and Dobrunz, 2015**; **Bhatia et al., 2019**; **Grangeray-Vilmint et al., 2018**). Consequently, the MC hypo-excitability in *Fmr1* KO mice effectively reduces excitatory drive onto local interneurons, which in turn reduces local inhibitory drive to granule cells, ultimately leading to abnormal dentate information processing and exaggerated granule cell output. Given that granule cells normally fire sparsely during information processing, this increased AP output in granule cells is reminiscent of abnormal CA3-to-CA1 information transmission in the *Fmr1* KO mice, in which a large amount of "noise' that is normally filtered out during synaptic processing is being transmitted indiscriminately due to FMRP loss (**Deng et al., 2013**).

We note the discrepancy in the changes of inhibitory drives among different cell types, that is, sIPSC frequency was increased in MCs, showed no change in granule cells and a decrease in interneurons, which cannot be explained by a simple change in excitability of any one cell type in dentate gyrus. Rather, these distinct changes might indicate a compensatory circuit reorganization of the interneuron axonal terminals onto these three targets in the absence of FMRP. Similarly, the observed alterations in excitatory drive onto interneurons, including both mEPSC and sEPSC frequencies, suggest changes in the excitatory synapse number and/or function. Together with alterations in inhibitory drives, these changes may reflect compensatory reorganization of both excitatory and inhibitory connections, including MC synapses. Indeed, FMRP plays a crucial role in synaptogenesis and synaptic refinement (**Booker et al., 2019**; **Doll and Broadie, 2014**; **Kennedy et al., 2020**; **Sears and Broadie, 2017**), and loss of FMRP is well known to cause changes in synaptic connectivity (**Gibson et al., 2008**; **Kennedy et al., 2020**; **Patel et al., 2013**; **Patel et al., 2014**; **Tessier and Broadie, 2009**). Such circuit reorganization can explain the balanced E/I drive onto granule cells in *Fmr1* KO mice we observed in the

basal state, which can result from reorganization of inhibitory axonal terminals in favor of increased contacts onto granule cells and MCs, and reduced contacts onto other interneurons, as a compensation for the decreased MC-derived excitatory drive. In line with this notion, it has been reported that dentate circuit reorganization occurs after status epilepticus, which can increase inhibitory drive onto granule cells and function as a homeostatic compensation (*Butler et al., 2022*). Indeed, many of the cellular and synaptic changes in the *Fmr1* KO mice are antagonistic and mitigating circuit dysfunction, and thus compensate for the primary defects (*Domanski et al., 2019*). However, even though the homeostatic circuit compensation is able to maintain the basal E/I input balance in granule cells of *Fmr1* KO mice, this compensation was insufficient to maintain dynamic E/I balance during active states, such as during network activity evoked by PP stimulation. Consequently, the dentate output was exaggerated during the physiologically relevant theta–gamma coupling activity in *Fmr1* KO mice. Indeed, as discussed above, E/I balance is dynamically tuned to rapidly adjust during circuit computations (*Bartley and Dobrunz, 2015*; *Bhatia et al., 2019*; *Grangeray-Vilmint et al., 2018*), which is thought to represent a fundamental mechanism underlying efficient neural coding (*Zhou and Yu, 2018*). Considering that the theta–gamma oscillation coupling plays a critical role in dentate information processing (*Bott et al., 2016*; *Leutgeb et al., 2007*; *Lisman and Jensen, 2013*; *Mizuseki et al., 2009*; *Neves et al., 2022*; *Pernía-Andrade and Jonas, 2014*), our findings indicate that MC-driven E/I imbalance is a critical defect contributing to dentate dysfunction in FXS. Moreover, by projecting axons both ipsi- and contralaterally along the longitudinal axis of the dentate gyrus, MCs are uniquely positioned to influence dentate activity widely across lamellae (*Jinde et al., 2013*; *Scharfman, 2016*; *Scharfman, 2018*). This wide-reaching control of dentate activity makes MC hypo-excitability in *Fmr1* KO mice a core defect in the dysregulation of circuit E/I dynamics and information processing.

Most critical discovery in this regard is our finding that circuit-wide inhibition of Kv7 was sufficient to re-normalize E/I balance in the dentate circuit despite the irreconcilable hypo-excitable state of MCs and hyper-excitable state of granule cells in the *Fmr1* KO mice. Specifically, XE991 corrected MC hypo-excitability via direct effect on Kv7 channels in MCs and normalized granule cell excitability through dentate three-synapse feedback pathway in *Fmr1* KO mice. Moreover, inhibition of Kv7 channels was also sufficient to restore the dentate output during the physiologically relevant theta–gamma coupling activity in KO mice. Although inhibition of K$^+$ channels is typically expected to increase circuit excitability, the key role of MCs as the main excitatory drive for this three-synapse feedback inhibition resulted in the overall suppression of circuit hyperexcitability in the dentate circuit of *Fmr1* KO mice. This effect arises because the excessive Kv7 function is present selectively in MCs of *Fmr1* KO mice. As a result, inhibition of Kv7 had a stronger effect on enhancing MC-dependent feedback inhibition onto granule cells than directly increasing granule cell excitability in *Fmr1* KO mice. Notably, the applicability of a circuit-wide approach as a potential treatment in vivo will require extensive future behavioral analyses, which are beyond the scope of this study.

Taken together, these findings provide a proof-of-principle demonstration that a circuit-based intervention can normalize dynamic E/I balance and restore dentate circuit output in vitro. Thus, circuit-based interventions can represent a potential therapeutic strategy to correct dentate dysfunction in this disorder.

## Materials and methods

### Key resources table

| Reagent type (species) or resource | Designation | Source or reference | Identifiers | Additional information |
|---|---|---|---|---|
| Strain, strain background (mouse, *FVB,* male) | *Fmr1* KO mouse (FVB.129P2-*Pde6b$^+$ Tyrc$^{c-ch}$ Fmr1$^{tm1Cgr}$*/J) | The Jackson Laboratory | 004624 | |

*Continued on next page*

*Continued*

| Reagent type (species) or resource | Designation | Source or reference | Identifiers | Additional information |
|---|---|---|---|---|
| Strain, strain background (mouse, *FVB*, male) | Control for *Fmr1* KO mouse (FVB.129P2-*Pde6b*$^+$ *Tyr*$^{c\text{-}ch}$/AntJ) | The Jackson Laboratory | 004828 | |
| Antibody | Anti-α tubulin (rabbit polyclonal) | Abcam | Cat# ab18251, RRID:AB_2210057 | WB (1:20,000) |
| Antibody | Anti-FMRP (rabbit polyclonal) | Cell Science Technology | Cat# 4317, RRID:AB_1903978 | WB (1:1000) |
| Antibody | Anti-Kv7.2 (rabbit polyclonal) | Alomone Labs | Cat# APC-050, RRID:AB_2040101 | WB (1:500) |
| Antibody | Anti-Kv7.3 (rabbit polyclonal) | Alomone Labs | Cat# APC-051, RRID:AB_2040103 | WB (1:500) |
| Chemical compound, drug | (2S)–3-[[(1S)–1-(3,4-Dichlorophenyl)ethyl]amino-2-hydroxypropyl](phenylmethyl)phosphinic acid hydrochloride (CGP55845) | Tocris Bioscience | 1248 | |
| Chemical compound, drug | 4-(2-Hydroxyethyl)piperazine-1-ethanesulfonic acid, N-(2-Hydroxyethyl)piperazine-N′-(2-ethanesulfonic acid) (HEPES) | MilliporeSigma | H3375 | |
| Chemical compound, drug | 4,5,6,7-Tetrahydroisoxazolo[5,4c]pyridin-3-ol hydrochloride (THIP) | Tocris Bioscience | 0807 | |
| Chemical compound, drug | 6,7-Dinitroquinoxaline-2,3-dione(DNQX) | Tocris Bioscience | 2312 | |
| Chemical compound, drug | Adenosine 5'-triphosphate disodium (Na$_2$-ATP) | MilliporeSigma | A1852 | |
| Chemical compound, drug | Adenosine 5'-triphosphate magnesium (Mg-ATP) | MilliporeSigma | A9187 | |
| Chemical compound, drug | Guanosine 5'-triphosphate sodium (Na-GTP) | MilliporeSigma | G8877 | |
| Chemical compound, drug | Horseradish peroxidase (rabbit IgG conjugated) | Thermo Fisher | 656120 | |
| Chemical compound, drug | Picrotoxin (PTX) | Tocris Bioscience | 1128 | |
| Chemical compound, drug | QX-314 | MilliporeSigma | 552233 | |
| Chemical compound, drug | SR 95531 hydrobromide (Gabazine, GBZ) | Tocris Bioscience | 1262 | |
| Chemical compound, drug | Tetrodotoxin (TTX) | Tocris Bioscience | 1069 | |
| Software, algorithm | LabView | National Instrument | LabView 8.6 | |
| Software, algorithm | MATLAB | MathWorks | MATLAB 2012b | |
| Software, algorithm | Origin | Origin Labs | Origin 8.5 | |
| Software, algorithm | Mini Analysis | Synaptosoft Inc | Version 6.0.3 | |

## Animals and slice preparation

*Fmr1* KO (FVB.129P2-Pde6b$^+$ Tyr$^{c\text{-}ch}$ Fmr1$^{tm1Cgr}$/J; stock #004624) and WT control mice (FVB.129P2-Pde6b$^+$Tyr$^{c\text{-}ch}$/AntJ; stock #004828) were obtained from The Jackson Laboratory. Slices were prepared as previously described (*Deng et al., 2022*). In brief, male 21—23-day-old mice were used. After being deeply anesthetized with $CO_2$, mice were decapitated and their brains were dissected out in ice-cold saline containing the following (in mm): 130 NaCl, 24 NaHCO$_3$, 3.5 KCl, 1.25 NaH$_2$PO$_4$, 0.5 CaCl$_2$, 5.0 MgCl$_2$, and 10 glucose, pH 7.4 (saturated with 95% O$_2$ and 5% CO$_2$). Horizontal hippocampal slices (350 μm) were cut using a vibrating microtome (Leica VT1100S) (*Deng et al., 2022*).

Slices were initially incubated in the above solution at 35°C for 1 hr for recovery and then kept at room temperature (~23°C) until use. All animal procedures were in compliance with the US National Institutes of Health Guide for the Care and Use of Laboratory Animals, and conformed to Washington University Animal Studies Committee guidelines.

## AP recording

AP recordings using a Multiclamp 700B amplifier (Molecular Devices) were made from MCs, interneurons or granule cells with infrared video microscopy and differential interference contrast optics (Olympus BX51WI). All of the recordings were conducted at near-physiological temperature (33–34°C). The recording electrodes were filled with the following in this study (unless stated otherwise) (in mM): 130 K-gluconate, 10 KCl, 0.1 EGTA, 2 $MgCl_2$, 2 $ATPNa_2$, 0.4 GTPNa, and 10 HEPES (pH 7.3). The extracellular solution contained (in mM) 125 NaCl, 24 $NaHCO_3$, 3.5 KCl, 1.25 $NaH_2PO_4$, 2 $CaCl_2$, 1 $MgCl_2$, and 10 glucose, pH 7.4 (saturated with 95% $O_2$ and 5% $CO_2$).

MCs were identified by their location (hilus), characteristic morphology (size and shape), and electrophysiological properties, including RMP (~–65 mV), cell capacitance (>100 pF), input resistance (~150 MΩ at RMP), sEPSCs with high frequency and large amplitude (>60 pA), and AP with relatively small afterhyperpolarization (~–6 mV) (*Bui et al., 2018*; *Chancey et al., 2014*; *Jinde et al., 2012*; *Scharfman and Myers, 2012*; *Wang et al., 2021*). For determination of excitability, MC membrane potential was set to given potentials (–64 to –55 mV with 1 mV step) through amplifier's function of automatic current injection. The number of APs within 20 s was averaged from 4 to 5 trials for each cell. For measurements of AP parameters, we employed a ramp current injection (0.15 pA/ms) (*Deng et al., 2019*; *Deng et al., 2022*) with a hyperpolarizing onset to ensure maximal $Na^+$ channel availability before the first AP. All AP parameters (except number of APs) were determined only from the first APs of ramp-evoked AP trains to avoid the influence from cumulatively inactivating voltage-gated channels. AP threshold was defined as the voltage at the voltage trace turning point, corresponding to the first peak of third-order derivative of AP trace (*Deng et al., 2022*). The rheobase was defined as the ramp current amplitude at the time point of threshold; rheobase charge transfer was determined by the integration of time and input current. The maximum rising rate was determined as the peak of the first-order derivative of AP trace. AP duration was measured at the level of AP trace crossing –10 mV. The AP amplitude was measured from the threshold point to the AP peak because the membrane potential was continuously ramp-up with time and the threshold point is the most reliable point before an AP in this setting. The rise and fall time was the interval of 10–90% amplitude during AP upstroke and downstroke, respectively.

Hilar interneurons were easily distinguished from MCs according to the differences in morphology and electrophysiological properties (*Lübke et al., 1998*) (relative to MCs, interneurons have more depolarizing RMP [~–55 mV], smaller capacitance [<60 pF], higher input resistance [usually >300 MΩ at RMP], and larger afterhyperpolarization [lower than –15 mV]). AP firing pattern in response to a 100 pA depolarizing step current for 600 ms was used to classify interneurons into three types (*Figure 5—figure supplement 1A*).

Granule cells were identified by the location (granule cell layer) and characteristic electrophysiological properties (relative to interneurons) (*Deng et al., 2022*): very hyperpolarizing RMP (~ –80 mV), very low sEPSC frequency (<20 events/min), and smaller capacitance (~20 pF). Cells that could not be definitively classified into the three categories above were not included in further analyses. To avoid recording from newly generated immature granule cells, we used cells located at the outer regions of the granule cell layer (*Deng et al., 2022*). For step-current-evoked AP output in granule cells, we used multistep current (50–75 pA with an increase of 5 pA/step for 600 ms) to evoke APs from granule cells. The RMP was set to –80 mV by constant current injection (if necessary) for better comparison among cells. The number of APs was averaged over 5–8 trials in each cell. For PP stimulation-evoked AP in granule cells, the stimulation protocol consisted of a train of 5 Hz; the stimulation electrodes positioned in the middle molecular layer of dentate gyrus to stimulate medial PP. The RMP of granule cells was set to –70 mV for facilitating AP firing and the stimulation intensity was adjusted so that the AP probability was ~0.5 (under 5 Hz stimulation). A burst of gamma stimulation (five stimuli at 50 Hz, 200 ms before the first stimulus of 5 Hz train) was used to evaluate gamma stimulation-induced suppression of granule cell output in response to PP theta stimulation. Granule cell output was expressed as the AP probability. When PP stimulation failed to evoke APs,

we measured the amplitude of EPSPs, which was defined as the voltage differences between individual EPSP peaks and the initial baseline before gamma-stimulation (set to be –70 mV). For better comparison, the EPSP amplitudes were normalized to their own controls. AP probability was calculated from at least 10 trials in each cell.

## Measurement of RMP, capacitance, and input resistance

RMP was measured immediately after whole-cell formation. Cell capacitance is determined by the amplifier's auto whole-cell compensation function with slightly manual adjustment to optimize the measurement if needed. For input resistance, assessment was performed using the canonical method, in which the voltage response to a negative current injection (–60 pA for 500 ms) at RMP was used to calculate input resistance. For measurement of input resistance around threshold levels, we pharmacologically isolated MCs using blockers against both glutamate and GABA receptors (in µM, 10 NMDA, 50 APV, 10 MPEP, 5 gabazine, and 2 CGP55845) to isolate MCs from the dentate circuit. TTX (1 µM) and $CdCl_2$ (10 µM) were also used to block $Na^+$ and $Ca^{2+}$ channels, respectively. Under these conditions, a positive current injection (+60 pA for 500 ms) at –45 mV (set by constant current injection) was used to evoke voltage response, which was then used to calculate input resistance.

## Determination of changes in holding current and membrane potential

Holding current was recorded by holding cells at –45 mV in pharmacologically isolated MCs, and in the presence of TTX (1 µM) and 10 µM $CdCl_2$. Changes in holding current were the differences in holding current before and during XE991 (10 µM). For determining changes in membrane potentials in response to Kv7 inhibition, membrane potentials were initially set at –45 mV (by constant current injection). Under these conditions, the differences in membrane potentials before and during XE991 (10 µM) were calculated as the changes in membrane potential.

## Kv7 current recording

We used a depolarizing voltage ramp (–95 to +5 mV, 0.02 mV/ms) to evoke Kv7 current from MCs using the same internal and external solutions as those in holding current recordings. Cell capacitance was compensated. Series resistance compensation was enabled with 80–90% correction and 16 µs lag. Kv7 current was isolated by subtracting current in 10 µM XE991 from that before XE991. The I-V curves were constructed from ramp-evoked Kv7 currents every 5 mV and normalized to respective cell capacitances (mean current value over 0.01 mV intervals from averages of 4–5 trials for each cell to approximate quasi-steady-state current).

## Recordings of spontaneous and miniature postsynaptic currents

sEPSCs were recorded from hilar MCs (holding at –65 mV) and interneurons (at –60 mV). The pipette solution was the same as that used in AP recording, except that QX-314 (1 mM) was included in the pipette solution to block possible action current. The bath solution was supplemented with gabazine (5 µM) to block $GABA_AR$ responses. The solutions used for recording of mEPSCs were the same as those for sEPSCs, except that TTX (1 µM) was included in the bath solution to block AP-dependent responses.

For recording of sIPSCs, the recording pipette solution contained (in mM) 130 CsCl, 2 $MgCl_2$, 4 Mg-ATP, 0.3 Na-GTP, 10 HEPES, and 0.1 EGTA (pH 7.3). The bath solution was supplemented with APV (50 µM) and DNQX (10 µM) to block responses of ionotropic glutamate receptors. Note that IPSC is downgoing signal in this high chloride internal solution. For mIPSCs' recording, TTX was added in the bath solution.

For simultaneous recording of sEPSCs and sIPSCs from granule cells, the recording pipette solution contained (in mM) 135 K-gluconate, 2 $MgCl_2$, 0.1 $CaCl_2$, 2 MgATP, 0.3 NaGTP, 4 $Na_2$-phosphocreatine, 0.2 EGTA, and 10 HEPES (pH 7.3). External solution was the same as AP recordings (without any blockers, unless stated otherwise). The membrane potential was held at –40 mV. Under this condition, EPSC is downgoing signal (inward current) and IPSC upgoing signal (outward current). The events detection threshold was 10 pA for sEPSC and sIPSC detection; the automatic detection was visually verified.

## Recording of cPSC and isolation of underlying EPSC and IPSC

The cPSC was recorded from granule cells by stimulation of medial PP at 0.2 Hz and using the same pipette solution as that of simultaneous recording of sEPSCs and sIPSCs, except that QX314 (1 mM) was included to block action current. Granule cells were held at –45 mV, which is an intermediate potential between the excitatory and inhibitory reversal potentials ensuring comparable driving force for excitatory and inhibitory conductances. The PP-stimulation-evoked cPSC had an initial downward excitatory component followed by an upward inhibitory component (*Figure 8—figure supplement 1A*). The cPSC excitation window was defined as the full duration of excitatory component (*Figure 8—figure supplement 1C*). At the end of each recording, the pure EPSC (*Figure 8—figure supplement 1B*) was recorded by adding GABA$_A$ blocker gabazine (5 µM) and keeping the same stimulation intensity, which was used to create an EPSC template (average of at least 20 uncontaminated EPSCs) for each cell. The EPSC template was then repeatedly scaled to each data point of cPSC 'approximating segment' to obtain a set of scaled EPSCs (*Figure 8—figure supplement 1C*). All scaled EPSCs were then averaged to approximate an underlying EPSC for a given cPSC (*Figure 8—figure supplement 1C*). The 'approximating segment' of cPSC was defined as 25–65% height of cPSC excitatory component, but not beyond 2.5 ms after stimulation. The underlying IPSC was isolated by subtracting the underlying EPSC from corresponding cPSC (*Figure 8—figure supplement 1D*). For better comparison, we normalized the cPSC and underlying IPSC to their respective underlying EPSC, which reflects the PP stimulation intensity (*Figure 8—figure supplement 1E*). The underlying EPSC and IPSC were then used to estimate E/I ratio by their peak amplitudes or charge transfer (amplitude-time integration within 100 ms). Data for each cell were averages from 20 to 25 uncontaminated cPSC events.

## Western blotting

Whole brains or dentate gyrus regions (isolated from the brain slices) were lysed in 2% sodium dodecyl sulfonate (SDS) with protease and phosphatase inhibitors (Roche Applied Sciences) and manually homogenized. Protein concentration was determined by DC protein assay (Bio-Rad Laboratories) against bovine serum albumin standards. Then, 50 µg total protein was run on NuPage 4–12% Bis-Tris polyacrylamide gels (Life Technologies) and transferred to nitrocellulose membranes. Membranes were blotted with antibodies directed against the following proteins: actin, FMRP, KCNQ2, KCNQ3, rabbit IgG conjugated to horseradish peroxidase. Blots were developed with SuperSignal West Dura (Thermo Fisher) and imaged with a ChemiDoc MP imaging system (Bio-Rad Laboratories).

## Statistical analysis

The data were analyzed in MATLAB, except for the postsynaptic currents (sEPSC, mEPSC, sIPSC, and mIPSC) that were analyzed by MiniAnalysis. All figures were made in Origin or MATLAB. Data are presented as mean ± SEM. Student's *t*-test, one-way ANOVA, Kolmogorov–Smirnov (K-S) test, or chi-square test were used for statistical analysis as appropriate. Significance was set as $p < 0.05$. The n is the number of cells tested in electrophysiological experiments, which was from at least three different mice for each condition. The N in western blotting is the number of animals used. All statistical values can be found in *Supplementary file 1*.

# Acknowledgements

This work was supported in part by NIH grant R35 NS111596 to VAK and R35 NS122260 to VC. The schematic illustration of *Figure 1A* was created by Yuhan Deng.

# Additional information

### Funding

| Funder | Grant reference number | Author |
|---|---|---|
| National Institute of Neurological Disorders and Stroke | R35 NS111596 | Vitaly A Klyachko |

| Funder | Grant reference number | Author |
|---|---|---|
| National Institute of Neurological Disorders and Stroke | R35 NS122260 | Valeria Cavalli |

The funders had no role in study design, data collection and interpretation, or the decision to submit the work for publication.

## Author contributions

Pan-Yue Deng, Conceptualization, Data curation, Formal analysis, Validation, Investigation, Visualization, Writing – original draft, Writing – review and editing; Ajeet Kumar, Data curation, Formal analysis, Investigation, Writing – review and editing; Valeria Cavalli, Conceptualization, Supervision, Funding acquisition, Writing – original draft, Project administration, Writing – review and editing; Vitaly A Klyachko, Conceptualization, Resources, Data curation, Supervision, Funding acquisition, Writing – original draft, Project administration, Writing – review and editing

## Author ORCIDs

Pan-Yue Deng ⬤ http://orcid.org/0000-0002-9577-9074
Valeria Cavalli ⬤ http://orcid.org/0000-0001-9978-050X
Vitaly A Klyachko ⬤ https://orcid.org/0000-0003-3449-243X

## Ethics

All animal procedures were in compliance with the US National Institutes of Health Guide for the Care and Use of Laboratory Animals, and approved by Washington University Animal Studies Committee (IACUC protocol #23-0194).

Reviewer #1 (Public Review): https://doi.org/10.7554/eLife.92563.3.sa1
Reviewer #2 (Public Review): https://doi.org/10.7554/eLife.92563.3.sa2
Reviewer #3 (Public Review): https://doi.org/10.7554/eLife.92563.3.sa3
Author Response https://doi.org/10.7554/eLife.92563.3.sa4

# Additional files

## Supplementary files

• Supplementary file 1. Table containing statistical information for all figures.
• MDAR checklist

## Data availability

All data generated and analyzed in this study are included in the manuscript and supporting files.

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
