## [Editor Report · eLife assessment]

This is a **fundamental** work that significantly advances our understanding of the role of mossy cells in the dentate gyrus in fragile X syndrome. The carefully designed and executed extensive series of experiments provide **compelling** evidence that changes in their excitability occur due to upregulation of Kv7 currents. The study unveils the underlying mechanisms of the disease, and therefore the work will be of interest to neuroscientists working on various aspects of fragile X pathology. In addition, it also provides insights into how neuronal activity is balanced in networks through diverse cellular mechanisms.

---

## [Referee Report · Reviewer #1 (Public Review)]

Summary:

In this work the authors provide evidence to show that an increase in Kv7 channels in hilar mossy cells of Fmr1 knock out mice results in a marked decrease in their excitability. The reduction in excitatory drive onto local hilar interneurons produces an increased excitation/inhibition ratio in granule cells. Inhibiting Kv7 channels can help normalize the excitatory drive in this circuit, suggesting that they may represent a viable target for targeted therapeutics for fragile-x syndrome.

Strengths:

The work is supported by a compelling and thorough set of electrophysiological studies. The authors do an excellent job of analysing their data and present a very complete data set.

Weaknesses:

There are no significant weaknesses in the experimental work, however the complexity of the data presentation and the lack of a schematic showing the organizational framework of this circuit make the data less accessible to non-experts in the field. I highly encourage a graphical abstract and network diagram to help individuals understand the implications of this work.

The work is important as it identifies a unique regional and cell specific abnormality in Fmr1 KO mice, showing how the loss of one gene can result in region specific changes in brain circuits.

---

## [Referee Report · Reviewer #2 (Public Review)]

Summary:

Deng et al. investigate, for the first time to my knowledge, the role that hippocampal dentate gyrus mossy cells play in Fragile X Syndrome. They provide compelling evidence that, in slice preparations from Fmr1 knockout mice, mossy cells are hypoactive due to increased Kv7 function whereas granule cells are hyperactive compared to slices from wild-type mice. They provide strong evidence that weakness of mossy cell-interneuron connections contribute to granule cell hyperexcitability, despite converse adaptations to mossy cell inputs. The authors show that application of the Kv7 inhibitor XE991 is able to rescue granule cell hyperexcitability back to wild-type baseline, supporting the overall conclusion that inhibition of Kv7 in the dentate may be a potential therapeutic approach for Fragile X Syndrome.

Strengths:

Thorough electrophysiological characterization of mossy cells in Fmr1 knockout mice, a novel finding.

Their electrophysiological approach is quite rigorous: patched different neuron types (GC, MC, INs) one at a time within the dentate gyrus in FMR1 KO and WT, with and without 'circuit blockade' by pharmacologically inhibiting neurotransmission. This allows the most detailed characterization possible of passive membrane/intrinsic cell differences in dentate gyrus of Fmr1 knockout mice.

Provide several examples showing the use of Kv7 inhibitor XE991 is able to rescue excitability of granule cell circuit in Fmr1 knockout mice (AP firing in intact circuit, postsynaptic current recordings, theta-gamma coupling stimulation)

Weaknesses:

Previously identified weaknesses have been addressed.

---

## [Referee Report · Reviewer #3 (Public Review)]

The first part of the review was prepared after the first submission of the paper. After this, the authors made several changes in the manuscript. These changes are assessed at the end of the review.

First part:

The paper by Deng, Kumar, Cavalli, Klyachko describe that, unlike in other cell types, loss of Fmr1 decreases the excitability of hippocampal mossy cells due to up-regulation of Kv7 currents. They also show evidence that while muting mossy cells appears to be a compensatory mechanism, it contribute to the higher activity of the dentate gyrus, because the removal of mossy cell output alleviate the inhibition of dentate principal cells. This may be important for the patho-mechanism in Fragile X syndrome caused by the loss of Fmr1.

These experiments were carefully designed, and the results are presented ‎in a very logical, insightful and self-explanatory way. Therefore, this paper represents strong evidences for the claims of the authors. In the current state of the manuscript there are only a few points that need additional explanation.

One of the results, that is shown in the supplementary dataset, does not fit to the main conclusions. Changes in the mEPSC frequency suggest that in addition to the proposed network effects, there are additional changes in the synaptic machinery or synapse number that are independent of the actual activity of the neurons. Since the differences of the mEPSC and sEPSC frequencies are similar and because only the latter can signal network effects, while the former is typically interpreted as a presynaptic change, it cannot be claimed that sEPSC frequency changes are due to the hypo-excitability of mossy cells.

An apparent technical issue may imply a second weak point in the interpretation of the results. Because the IPSCs in the PP stimulation experiments (Fig8) start within a few milliseconds, it is unlikely that its first ‎components originate from the PP-GC-MC-IN feedforward inhibitory circuit. The involvement of this circuit and MCs in the Kv7-dependent excitability changes is the main implication of the results of this paper. But this feedforward inhibition requires three consecutive synaptic steps and EPSP-AP couplings, each of them lasting for at least 1ms + 2-5ms. Therefore, the inhibition via the PP-GC-MC-IN circuit can be only seen from 10-20ms after PP stimulation. The earlier components of the cPSCs should originate from other circuit elements that are not related to the rest of the paper. Therefore, more isolated measurements on the cPSC recordings are needed ‎which consider only the later phase of the IPSCs. This can be either a measurement on the decay phase or a pharmacological manipulation that selectively enhance/inhibit a specific component of the proposed circuit.

I suggest refraining from the conclusions saying "‎MCs provide at least ~51% of the excitatory drive onto interneurons in WT and ~41% in KO mice", because too many factors (eg. IN celll types, slice condition, synaptic reliability) are not accounted for these actual numbers, and these values are not necessary for the general observation of the paper.

There are additional minor issues about the presentation of the results that are explained in the private recommendation for the Authors.

Review after the revision:

The authors accepted my suggestions and made changes in the manuscript to address my point about the interpretation of the mEPSC changes.

The second point was related to the interpretation of the stimulation evoked multisynaptic compound responses. Specifically, the IPSC components in the PP stimulation experiments start within a few milliseconds, and I pointed out that it is unlikely that its first ‎components originate from the PP-GC-MC-IN feedforward inhibitory circuit. The authors provided strong arguments for the interpretation of these compound responses in their reply and the conclusions are consistent with these complex results.

Additional minor issues were fully addressed.

I still think that this is a strong paper that provides new insights into the mechanisms of Fragile X syndrome at the level of single neurons and local network. The extensive series of experiments convincingly support the main findings that in addition to contributing to the underlying mechanisms of this disease also highlight how delicately neuronal activity is balanced even in constrained conditions.

---

## [Author Response]

The following is the authors’ response to the original reviews.

**Public Reviews:**

**Reviewer #1 (Public Review):**
Summary:In this work, the authors provide evidence to show that an increase in Kv7 channels in hilar mossy cells of Fmr1 knock out mice results in a marked decrease in their excitability. The reduction in excitatory drive onto local hilar interneurons produces an increased excitation/inhibition ratio in granule cells. Inhibiting Kv7 channels can help normalize the excitatory drive in this circuit, suggesting that they may represent a viable target for targeted therapeutics for fragile-x syndrome.Strengths:The work is supported by a compelling and thorough set of electrophysiological studies. The authors do an excellent job of analysing their data and present a very complete data set.

We thank the Reviewer for the positive comments.

Weaknesses:There are no significant weaknesses in the experimental work, however the complexity of the data presentation and the lack of a schematic showing the organizational framework of this circuit make the data less accessible to non-experts in the field. I highly encourage a graphical abstract and network diagram to help individuals understand the implications of this work.

We thank the Reviewer for the suggestion, and added a schematic of the dentate network organization (Figure 1A).

The work is important as it identifies a unique regional and cell-specific abnormality in Fmr1 KO mice, showing how the loss of one gene can result in region-specific changes in brain circuits.
**Reviewer #2 (Public Review):**
Summary:Deng et al. investigate, for the first time to my knowledge, the role that hippocampal dentate gyrus mossy cells play in Fragile X Syndrome. They provide strong evidence that, in slice preparations from Fmr1 knockout mice, mossy cells are hypoactive due to increased Kv7 function whereas granule cells are hyperactive compared to slices from wild-type mice. They provide indirect evidence that the weakness of mossy cell-interneuron connections contributes to granule cell hyperexcitability, despite converse adaptations to mossy cell inputs. The authors show that application of the Kv7 inhibitor XE991 is able to rescue granule cell hyperexcitability back to wild-type baseline, supporting the overall conclusion that inhibition of Kv7 in the dentate may be a potential therapeutic approach for Fragile X Syndrome. However, any claims regarding specific circuit-based intervention or analysis are limited by the exclusively pharmacological approach of the manipulations.Strengths:Thorough electrophysiological characterization of mossy cells in Fmr1 knockout mice, a novel finding.Their electrophysiological approach is quite rigorous: patched different neuron types (GC, MC, INs) one at a time within the dentate gyrus in FMR1 KO and WT, with and without 'circuit blockade' by pharmacologically inhibiting neurotransmission. This allows the most detailed characterization possible of passive membrane/intrinsic cell differences in the dentate gyrus of Fmr1 knockout mice.Provide several examples showing the use of Kv7 inhibitor XE991 is able to rescue excitability of granule cell circuit in Fmr1 knockout mice (AP firing in the intact circuit, postsynaptic current recordings, theta-gamma coupling stimulation).

We thank the Reviewer for the positive comments.

Weaknesses:The implications for these findings and the applicability of the potential treatment for the disorder in a whole animal are limited due to the fact that all experiments were done in slices.

We appreciate the Reviewer’s point and agree. To address this concern, we have revised the Discussion to state that “the applicability of a circuit-wide approach as a potential treatment in vivo will require extensive future behavioral analyses, which are beyond the scope of the current study”. We also now emphasize in Discussion that “these findings provide a proof-of-principle demonstration that a circuit-based intervention can normalize dynamic E/I balance and restore dentate circuit output in vitro”.

The authors' interpretation of the word 'circuit-based' is problematic - there are no truly circuit-specific manipulations in this study due to the reliance on pharmacology for their manipulations. While the application of the Kv7 inhibitor may have a predominant effect on the circuit through changes to mossy cell excitability, this manipulation would affect many other cells within the dentate and adjacent brain regions that connect to the dentate that express Kv7 as well.

We appreciate the reviewer’s point but would like to clarify that by using a term “circuit-based” we did not intend to imply that it is a “’circuit-specific” intervention. Our intended interpretation of the term ‘circuit-based’ stems from the following reasoning: the dentate circuit has two types of excitatory neurons which show opposite excitability defects in FXS mice, thus presenting an irreconcilable conflict to correct pharmacologically for each cell type individually. Instead, we sought an approach to correct the overall dentate circuit output, rather than to restore excitability defects of individual cell types. Notably, when we pharmacologically isolated granule cells from the circuit, inhibition of Kv7 failed to restore their excitability, suggesting that normalization of the dentate output depends on the circuit activity. Since we focused on correcting dentate output using such a circuit-dependent approach, we used the term ‘circuit-based intervention’ to emphasize this notion.

**Reviewer #3 (Public Review):**
The paper by Deng, Kumar, Cavalli, Klyachko describes that, unlike in other cell types, loss of Fmr1 decreases the excitability of hippocampal mossy cells due to up-regulation of Kv7 currents. They also show evidence that while muting mossy cells appears to be a compensatory mechanism, it contributes to the higher activity of the dentate gyrus, because the removal of mossy cell output alleviates the inhibition of dentate principal cells. This may be important for the patho-mechanism in Fragile X syndrome caused by the loss of Fmr1.These experiments were carefully designed, and the results are presented ‎in a very logical, insightful, and self-explanatory way. Therefore, this paper represents strong evidence for the claims of the authors. In the current state of the manuscript, there are only a few points that need additional explanation.

We thank the Reviewer for the positive comments.

One of the results, which is shown in the supplementary dataset, does not fit the main conclusions. Changes in the mEPSC frequency suggest that in addition to the proposed network effects, there are additional changes in the synaptic machinery or synapse number that are independent of the actual activity of the neurons. Since the differences of the mEPSC and sEPSC frequencies are similar and because only the latter can signal network effects, while the former is typically interpreted as a presynaptic change, it cannot be claimed that sEPSC frequency changes are due to the hypo-excitability of mossy cells.

We thank the Reviewer for this important point and agree. To address this concern, we now state in Results that “We note that changes in the excitatory drive onto interneurons include both mEPSC and sEPSC frequencies, which reflect not only potential deficits in excitability of their input cells, such as MCs, but also changes in synaptic connectivity/function, that may arise from homeostatic circuit reorganization/compensation (see Discussion)”.

We also now emphasize this point in Discussion by stating that “alterations in excitatory drives, including both mEPSC and sEPSC frequencies onto interneurons, suggest changes in the excitatory synapse number and/or function. Together with alterations in inhibitory drives these changes may reflect compensatory circuit reorganization of both excitatory and inhibitory connections, including mossy cell synapses”.

We also note in Discussion that “Such circuit reorganization can explain the balanced E/I drive onto granule cells in Fmr1 KO mice we observed in the basal state, which can result from reorganization of excitatory and inhibitory axonal terminals”.

Notably, our findings that Kv7 blocker acting by increasing MC excitability is sufficient to correct dentate output, supports the notion that hypo-excitability of mossy cells is a major factor contributing to dentate circuit E/I imbalance. This does not exclude the presence of additional mechanisms contributing to E/I imbalance, such as changes of synaptic connectivity or release machinery. To reflect this point, we revised the Results to temper the initial claim that “this analysis supports the notion that the hypo-excitability of MCs in Fmr1 KO mice caused (now replaced with “is a major factor contributing to”) the reduction of excitatory drive onto hilar interneurons, which ultimately results in reduced local inhibition”.

An apparent technical issue may imply a second weak point in the interpretation of the results. Because the IPSCs in the PP stimulation experiments (Fig 8) start within a few milliseconds, it is unlikely that its first ‎components originate from the PP-GC-MC-IN feedforward inhibitory circuit. The involvement of this circuit and MCs in the Kv7-dependent excitability changes is the main implication of the results of this paper. But this feedforward inhibition requires three consecutive synaptic steps and EPSP-AP couplings, each of them lasting for at least 1ms + 2-5ms. Therefore, the inhibition via the PP-GC-MC-IN circuit can be only seen from 10-20ms after PP stimulation. The earlier components of the cPSCs should originate from other circuit elements that are not related to the rest of the paper. Therefore, more isolated measurements on the cPSC recordings are needed ‎which consider only the later phase of the IPSCs. This can be either a measurement of the decay phase or a pharmacological manipulation that selectively enhances/inhibits a specific component of the proposed circuit.

We appreciate the Reviewer’s point. As we mentioned in Results: “The EPSP measured in granule cells in response to the PP stimulation integrates both excitatory and inhibitory synaptic inputs onto granule cells, including the direct synaptic input from the PP and all the PP stimulation-associated feedforward and feedback synaptic inputs. In other words, the EPSP in granule cells integrates all dentate circuit ‘operations’.” As the Reviewer pointed out, this is also the case in the measurements of cPSCs, which comprise all of PP stimulation-associated feedforward and feedback inhibition. We thank the Reviewer for the suggestion to isolate specific components of IPSC. However, we did not attempt to do it in this study for three reasons. First, activity of all of these circuit components likely overlaps extensively in time and it is difficult to identify the specific time point that can separate contributions from earlier canonical feed-forward and feed-back components from the contribution of the later MC-dependent PP-GC-MC-IN feed-forward component. Notably the tri-synapse PP-GC-MC-IN component differs temporarily from the canonical di-synaptic (PP-GC-IN) feed-back inhibition only by a single synaptic activation step, resulting in only a few milliseconds difference. Moreover, the temporal differences in the contributions of these components vary widely among different recordings making a uniform analysis very difficult. Second, we used three different metrics to assess E/I changes in cPSC measurements, which capture a wide range of temporal processes and their integration, including peak-to-peak measurements, the charge transfer, and the excitation window metrics. Third, the principal readout in our study was the overall dentate output (i.e., granule cell firing), which reflects the integration of all dentate circuit ‘operations’ thus making the overall cPSC measurements appropriate, in our view, for this readout.

I suggest refraining from the conclusions saying "‎MCs provide at least ~51% of the excitatory drive onto interneurons in WT and ~41% in KO mice", because too many factors (eg. IN cell types, slice condition, synaptic reliability) are not accounted for in these actual numbers, and these values are not necessary for the general observation of the paper.

We thank the reviewer for this suggestion, and have revised the manuscript accordingly.

There are additional minor issues about the presentation of the results.

We have carefully checked and corrected the minor errors that reviewer pointed out.

**Recommendations for the authors:**
Revisions that are considered essential for improved assessment regarding the strengths of support of the claims:Temper claims regarding circuit-based effectsTemper claims regarding very specific quantitative assessments of synaptic drivesDifferentiate between monosynaptic inputs and inputs arriving through multiple synaptic contacts with proper analytical techniques.

We appreciate these suggestions and have revised the manuscript to address the concerns raised by the reviewers.

**Reviewer #1 (Recommendations For The Authors):**
The authors do an outstanding job of reviewing and presenting all of their data. This is a paper I will recommend all of my trainees read, as it is an excellent example of a complete research project. While I am impressed with the effort involved, I also wondered if the complexity and thoroughness of their presentations could make the story less accessible to non-expert readers. My comments are simply intended to help them present a more coherent and succinct story to a wider audience, though I am not sure I really provide any meaningful changes. This is simply a very thorough and complete body of work that the authors should be commended for. After reading it I felt they had gone above and beyond what most authors would provide in terms of data to support their story, and thus I had no doubt that a change in Kv7 plays a role in changing the excitability of the network.

We thank the Reviewer for the positive comments and great suggestions. We have made numerous changes to present our work in a more coherent and succinct way, in part by re-plotting some of the figures, as well as by adding a schematic of the dentate circuit in Figure 1.

Figure 1. A visual of mossy cells and the local circuit they are studying would be a useful addition to Figure. 1. I also feel this is important for conveying the story of how hypo-excitability can impact the E/I of the network. I think it has to be more of a cell structure/circuit-based figure than is presented in Supplementary Figure 8.

We thank the reviewer for this suggestion. We have added a schematic of the dentate circuit with all major cell types involved in Figure 1A.

Figure 1. A, B, and C tell a coherent story and are easy to understand. The interpretation of the phase plot in D is harder to access. Perhaps having this as a separate figure and providing a clearer presentation of the way the phaseplot was created (see Figure 3 Bove et al., 2019, Neuroscience 418; DOI: 10.1016/j.neuroscience.2019.08.048)

We appreciate the Reviewer’s point and agree. In order to keep Figure 1 more concise and readable, we removed the phase plot in the revised version. This change did not negatively impact the result presentation because the primary aim of this plot was to visualize changes in voltage threshold in an alternative way, but it was already clearly shown by the ramp-evoked AP traces (revised Figure 1D, insert), and thus was not essential to show.

Figure 1 E-N might be better situated in a supplementary graph as the characteristics of the AP aren't changing.

We understand the Reviewer’s point, but we feel it would be better to keep all action potential metrics together in one figure, to show that only a specific subset of parameters was affected in Fmr1 KO mice.

Figure 2: (A-D) I am not sure having so many figures is required given the focus is on having a small change in Ir at one membrane potential. I do worry that the significance appears to be due to 2 cells with an IR of over 100 in the WT group and 2 with an IR of around 62 in the KO group. All other cells are between 75-100 in both groups. I also worry a bit bc in the literature IRs between 55 and 125 seem to be commonly reported by groups that do this work normally (Buzsacki, Westbrook, etc.). I would be cautious about making too much out of this result.

We thank the Reviewer for these comments. We have performed additional analyses of these data, as also suggested by Reviewer 3 (Point #1), and improved presentation of the data in Figure 2D-F by showing the effect of XE991 on increasing input resistance in WT vs KO. We also plotted other panels in a similar way to show the comparisons between WT and KO, as well as comparisons within genotype +/- XE991, which makes the results easy to follow. For more details, please also see the response to Reviewer 3, Point 1.

Figure 2D-E: As in the text, this result is really pointing towards there being a Kv7 issue. Worries about the data in D aside, I think these two figures alone tell a clearer story. Figure 3 on the other hand tells a story of the effects of blocking Kv7 on membrane potential. Is this central to the story the others are trying to tell?

We thank the reviewer for this point. We believe that Figure 2, Figure 3 and Figure 4—figure supplement 1 together provide strong and multifaceted evidence to support changes in Kv7 function in Fmr1 KO mossy cells.

Figure 3. This is an interesting finding that shows how detailed their analysis was. Showing that the change in holding current in KO animals is greater than in WT is the first solid piece of evidence that there is a change in Kv7 in these cells that affects their excitability.

We appreciate the reviewer’s comment. As mentioned above, we believe that Figure 2, Figure 3 and Figure 4—figure supplement 1 together provide strong and multifaceted evidence to support changes in Kv7 function in Fmr1 KO mossy cells.

Figures 4 and 5 provide additional detail to support the idea that Kv& changes by showing how the E/I ratio and spontaneous minis are shifted in KO animals.

We thank the Reviewer for the comments.

Figures 6-8 build a compelling story for the reduction in excitatory drive in mossy cells affecting the network dynamics in excitatory/inhibitory interactions in DG cells.

We appreciate the Reviewer’s comment.

**Reviewer #2 (Recommendations For The Authors):**
1. Other than location and characteristic morphology, the other parameters that were used to identify mossy cells and granule cells were also parameters used to find differences in cellular properties between wild-type and Fmr1 KO mice (RMP, sEPSC frequency, etc.), which would confound the results shown. The use of available transgenic mouse lines would provide for a more unbiased screen of these cells. Afterhyperpolarization was also used as a parameter while screening cells, yet none of the data on this measurement is shown.

We thank the reviewer for this point and agree that transgenic mouse lines provide a more unbiased way to identify various types of neurons. However, since the present study involves analyses of at least three different types of neurons, establishing multiple transgenic lines labeling different types of dentate neurons in the Fmr1 KO mouse model would be very time consuming and beyond the current resources of the lab. We would also like to clarify that the three types of dentate neurons are easily distinguished according to the large differences in location, morphology and basal electrophysiological properties, none of which were essential in defining differences between genotypes. Specifically, granule cells are located in the granule cell layer, have a small cell body (<10 μm), RMP around -80mV, capacitance ~20 pF, and infrequent sEPSCs (<20 events/min); mossy cells are located in the hilus, have a large cell body (>15 μm), RMP around -65 mV, capacitance >100 pF, and fast afterhyperpolarization less than -10 mV (WT –5.1 ± 0.7 mV, KO -5.8 ± 0.5 mV); interneurons are located in the hilus or border of granule cell layer, have a relative smaller cell body (10-15 μm), RMP around -55 mV, capacitance <60 pF, and afterhyperpolarization larger than -15 mV (WT -20.4 ± 1.3 mV, KO -19.8 ±1.4 mV). We note that the cells that could not be definitively classified into the three categories were not included in analyses, and we have now clarified this further in the Methods. To address the reviewer’s second concern regarding AHP, we now provided the corresponding values in the Methods.

1. A definitive way to test the cell-autonomous nature of the Kv7 changes would be to use female mice, who will have a mosaic of cells affected by the fragile X chromosome, and the Fmr1 KO cells could be engineered to express GFP to help identify them from wild-type cells.

We agree and appreciate this suggestion. This could be an interesting follow up study to further verify the cell-autonomous nature of Kv7 changes.

1. The authors heavily rely on XE991 as a selective Kv7 blocker. Is it blocking all Kv7 channels at the concentration used? If so, given the significant expression of Kv7 in the dentate as shown by Western blot, is it surprising that there is no effect of this inhibitor on wild-type slices in most cases?

We thank the reviewer for this important point. We used 10x of IC50 concentration in the present study, suggesting that more than 80% of Kv7 should be blocked. Notably, we observed several effects of XE991 in WT mice: it significantly increased input resistance (new Figure 2D-F), and strongly enhanced AP firing evoked by step depolarization (Figure 7E-H), although we did not observe effect of XE991 in WT in the analyses of spiking evoked by theta-gamma stimulation in Figure 8. However, this is not surprising. If a parameter we measured is predominately cell-autonomous (for example, input resistance), the effects of XE991 are easy to observe. However, if a parameter reflects integration of all dentate circuit operations (for example, AP probability in response to theta-gamma stimulation), it is difficult to detect the effect of XE991 in WT mice because the dentate circuit of WT mice has larger capability to maintain E/I balance in response to XE991.

1. E/I ratio is a helpful concept, and it is heavily relied upon in the results text, but statistically shaky, especially for sEPSC:sIPSCs since you are combining uncertainty in the sEPSC and sIPSC to make one very uncertain ratio that doesn't undergo any subsequent statistical confirmation (such as in Fig 4I).

We appreciate the reviewer’s point and apologize for the confusion in presentation of Fig 4I (and 5I), due to lack of detailed explanation. The E/I ratio shown in Figs. 4I (and 5I) is a single data-point estimate calculated from the mean values of independent sEPSC and sIPSC measurements (Figs. 4G-H and 5G-H, respectively). This ratio was used only as an estimate/illustration of the changes, rather than a precise determination of the shift in E/I balance. Because there is only one data-point for this ratio, statistical analysis is not possible. For this reason we performed extensive additional analyses in Figures 7 and 8, in which the EPSC and IPSC were measured from the same cells and at the same time to define the actual E/I ratio with the corresponding statistical analyses (i.e., a real matched and dynamic E/I ratio).

1. Is this mGlur2/CB1 specificity to PP/granule and MC axons, respectively, true in the Fmr1 KO mice? It is possible that mGluR2 and CB1 expression patterns are altered in FMR1 KO, thus the assumption used to isolate these distinct inputs may not hold true.

This is a very good point. We do assume that the specificity of Group II mGluR and CB1 is similar between Fmr1 KO and WT mice, but this is an assumption that we have not directly verified. However, our results in Figures 7 and 8 strongly support this assumption, because if it were not true, then our intervention would be unlikely to correct the excessive dentate output.

1. XE991 only normalized GC firing when other cells were not pharmacologically blocked. The authors suggest this means blockage of MC Kv7 reduces GC excitability back to normal...presumably by increasing MC  IN  GC firing. This is a conclusion from many indirect comparisons (comparing XE991 effect on GC with/without GABA and glutamate blockers; comparing MC firing rates with/without XE991, and using CB1 agonist versus mGluR2 agonist to say it is mossy cells that are mostly controlling INs) - a clincher experiment would be to acutely knockdown Kv7 in mossy cells specifically and measure GC and IN firing.

Thank you, this is a great suggestion. Indeed, as an expansion of this project, in the future studies we are planning to manipulate excitability of mossy cells through manipulating Kv7, or using chemogenetic or optogenetic approaches.

1. The reasoning behind the FMRP-Kv7 connection is quite weak, citing the paper Darnell 2011 as "translational target", but FMRP has myriad translational targets.

We agree, and attempted to define the mechanism of increased Kv7 function using co-immunoprecipitation approach, as well as immunostaining to look at cell-type specific expression changes. However, both of these approaches were difficult to interpret due to technical limitations of the available antibodies. We also note that “We did not further investigate the precise mechanisms underlying enhancement of Kv7 function in the absence of FMRP, since the present study primarily focuses on the functional consequences of abnormal cellular and circuit excitability”. To address this concern, we extensively discussed the potential mechanisms of FMRP-Kv7 connection, acknowledged in Discussion that “further studies will be needed to elucidate the precise mechanism responsible for the increased Kv7 function in Fmr1 KO mice”, and will continue to investigate it in the future studies.

1. The authors attempt to look for changes in Kv7 expression with Western blot, but since they hypothesize that Kv7 changes are mainly in the mossy cells, it is perhaps not surprising that they would not be able to see any changes when they look at dentate as a whole. Staining for Kv7 subunits to look at expression on a cellular level would be beneficial.

We appreciate the reviewer’s suggestion. We attempted to perform the suggested experiments using immunostaining for KCNQ2, KCNQ3 and KCNQ5 in different subtypes of dentate neurons. However, these experiments failed to produce interpretable results due to technical limitations of the available antibodies.

1. Is Kv7 localization or splice/composition different in FMR1 KO mice?

This is a very good point. As we mentioned in Point 8 above, we were not able to perform these experiments and do not have the answer at this point.

1. Regarding the 3 subtypes of interneurons in the dentate, the authors are pooling data based on similar intrinsic properties, but this conclusion may be affected by the low number of recorded neurons for the regular-spiking type. In addition, it is unclear whether these different interneuron types have differential circuit connectivity (most likely) which would make it imperative to keep circuit analysis for interneurons segregated into these cell types.

We appreciate the reviewer’s point. Indeed, these different interneuron types may have distinct circuit connectivity and contributions to circuit activity. However, identification of these 3 types of interneurons and determination of their respective functions is in itself a very extensive set of experiments which is beyond the scope of the current manuscript. We also note that the functional readout of circuit activity in our measurements was the AP firing and EPSPs evoked in granule cells by PP stimulation, which integrate all dentate circuit operations, including all of the feedforward and feedback loops which are mediated by all of these different types of interneurons. For simplicity, we thus pooled all interneuron data for the purposes of this study. But we fully agree that extensive future work is required to elucidate interneuron-type specific changes in Fmr1 KO mice and their contributions to the dentate circuit dysfunction.

1. To do statistics treating each cell individually, and therefore assuming each cell is independent of one another, is not correct. Two cells from the same mouse will be more similar than two cells from different mice, therefore they are not independent data points. Nested statistical methods (n cells from o slices from p mice) will be important in future work, as discussed by (Aarts et al., Nat. Neurosci. 2014).

We agree with the Reviewer’s point and appreciate this suggestion. In the present study, the cells tested in electrophysiological experiments were from at least 3 different mice for each condition, which help minimize this kind of errors.

**Reviewer #3 (Recommendations For The Authors):**
Is there a difference in the Rin at -45mV of the control cell after the application of XE991? This is important to appreciate whether the XE991-sensitive conductances contribute to the basal excitability of MCs. Furthermore, the statistical comparison of the Rin at -45mV of the FXS animals in the control solution and in the presence of XE991 would be also important‎. Actually, the most accurate measurement would be to show a difference in the acute Kv7-blockade between control and FXS animals, if that is possible with this blocker. Additionally, it would be also informative if the bar graphs in Fig.2 D & E were merged for this purpose, similarly as in the later figures.

We thank the Reviewer for this suggestion and agree. Following this suggestion, we have re-plotted the data in Figure 2 accordingly. Specifically, we now show that XE991 significantly increased input resistance in both WT and KO mossy cells, and the effect of XE991 on increasing input resistance was markedly larger in KO than WT mossy cells. For other figures, we have plotted data in a similar way to show the comparisons between WT and KO, as well as comparisons within genotype +/- XE991.

Because of the cell-to-cell variability of the voltage responses, it would be more informative and representative if the average of traces from all cells were shown in Fig.2 D & E.

We agree with the Reviewer’s point. For clarity of presentation, we presented the cell-to-cell variability of the data as scatter points of input resistance values in the bar graph (Figure 2E), together with the representative traces (Figure 2D). Plotting the average traces from all cells would result in a total of 30 traces for all the WT and KO mice, which is difficult to visually assess clearly.

On page 7, please clarify the recorded cell type in this sentence: "In ‎contrast, WIN markedly reduced the number of sEPSCs in both WT and KO mice...".

We thank the Reviewer for pointing out this omission and have clarified it in the revised version.

In Figures 6 C, F, and I, the title of the Y-axis should be normalized frequency. Please also correct the figure legend accordingly because the current sentence can be also interpreted as the absolute or total number of events that were compared, irrespective of the duration of the recordings.

We thank the Reviewer for this point and have corrected the revised version accordingly.